# CHARACTERIZING LONG-TAIL CATEGORIES ON GRAPHS VIA A THEORY-DRIVEN FRAMEWORK

## ABSTRACT

In the context of long-tailed classification on graphs, the vast majority of existing work primarily revolves around the development of model debiasing strategies, with the aim of mitigating class imbalances and enhancing overall performance. Despite the notable success, there is very limited literature that provides a theoretical tool for characterizing the behaviors of long-tail categories in graphs and gaining insight into generalization performance in real-world scenarios. To bridge this gap, we propose the first generalization bound for long-tail classification on graphs by formulating the problem in the fashion of multi-task learning, i.e., each task corresponds to the prediction of one particular category. Our theoretical results show that the generalization performance of long-tailed classification is dominated by the overall loss range and the total number of tasks. Building upon the theoretical findings, we propose a novel generic framework TAIL2LEARN for long-tailed classification on graphs. In particular, we start with a hierarchical task grouping module that allows us to assign related tasks into hypertasks and thus control the complexity of task space; then, we further design a balanced contrastive learning module to adaptively balance the gradients of both head and tail classes to control the loss range across all tasks in a unified fashion. Finally, extensive experiments demonstrate the effectiveness of TAIL2LEARN in characterizing long-tail categories on real graphs. We publish our data and code at `https://anonymous.4open.science/r/Tail2Learn-CE08/`.

## 1 INTRODUCTION

The graph provides a fundamental data structure for modeling a wide range of relational data, ranging from financial transaction networks (Wang et al., 2019; Dou et al., 2020) to social science (Fan et al., 2019). Graph Neural Networks (GNNs) have achieved outstanding performance on node classification tasks (Zhang et al., 2019; Abu-El-Haija et al., 2020) because of their ability to learn expressive representations from graphs. Despite the remarkable success, the performance of GNNs is mostly attributed to the high-quality and abundant annotated data (Yang et al., 2020; Garcia & Bruna, 2017; Hu et al., 2019; Kim et al., 2019).

Nevertheless, unlike many graph benchmark datasets developed in the lab environment, it is often the case that many high-stake domains naturally exhibit a long-tail distribution, i.e., a few

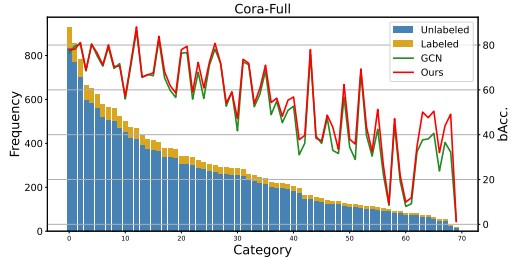

Figure 1: An illustrative figure of long-tail distribution in the collaboration network (Cora-Full), where the green and red curves show balanced accuracy (bAcc) (%) of GCN and TAIL2LEARN for node classification on each category. Blue and yellow bars represent the class frequency of unlabeled and labeled nodes.

head classes[1] (the majority classes) are well-studied with rich data, while the massive tail classes (the minority classes) are under-explored with scarce data. For example, in financial transaction networks, a few head classes correspond to the normal transaction types (e.g., credit card payment,

---

[1]In this paper, we use 'class' and 'category' interchangeably.

wire transfer), and the numerous tail classes can represent a variety of fraudulent transaction types (e.g., money laundering, synthetic identity transaction). Despite the rare occurrences of fraudulent transactions, detecting them can prove crucial (Singleton & Singleton, 2010; Akoglu et al., 2015). Another example is the collaboration network. As shown in Figure 1, the Cora-Full network (Bojchevski & Günnemann, 2018) is composed of 70 categories based on research areas, which follow a highly-imbalanced data distribution (e.g., 15 papers in the most niche area while 928 papers in the most popular area). The task complexity (data imbalance, massive classes) coupled with limited supervision bring enormous challenges to GNNs.

Important as it could be, there is limited literature that provides a theoretical grounding to characterize the behaviors of long-tail categories on graphs and understand the generalization performance in real environments. To bridge the gap, we provide insights and identify three fundamental challenges in the context of long-tail classification on graphs. First (***C1. Highly-skewed data distribution***), the data exhibits extremely skewed class memberships. Consequently, the head classes contribute more to the learning objective and can be better characterized by GNNs; the tail classes contribute less to the objective and thus suffer from higher systematic errors (Zhang et al., 2021). Second (***C2. Label scarcity***), due to the rarity and diversity of tail classes in nature, it is often more expensive and time-consuming to annotate tail classes rather than head classes (Pelleg & Moore, 2004). What is worse, training GNNs from scarce labels may result in representation disparity and inevitable errors (Zhou et al., 2019; Wang et al., 2021), which amplifies the difficulty of debiasing GNN from the highly-skewed data distribution. Third (***C3. Task complexity***), with the increasing number of categories, the difficulty of separating the margin (Hearst et al., 1998) of categories is dramatically increasing. There is a high risk of encountering overlapped regions between classes with low prediction confidence (Zhang & Zhou, 2013; Mittal et al., 2021). To deal with the long-tail categories, the existing literature mainly focuses on augmenting the observed graph (Zhao et al., 2021; Wu et al., 2021; Qu et al., 2021) or reweighting the category-wise loss functions (Yun et al., 2022; Shi et al., 2020). Despite the existing achievements, a natural research question is that: *can we further improve the overall performance by learning more knowledge from both head classes and tail classes?*

To answer the aforementioned question, we provide the first study on the generalization bound of long-tail classification. The key idea is to formulate the long-tail classification problem in the fashion of multi-task learning (Song et al., 2022b), i.e., each task corresponds to the prediction of one particular category. In particular, the generalization bound is in terms of the range of losses across all tasks and the total number of tasks. Building upon the theoretical findings, we propose TAIL2LEARN, a generic learning framework to characterize long-tail categories on graphs. Specifically, we utilize a hierarchical structure for task grouping to address C2 and C3, which assigns related tasks into hypertasks in order to control the complexity of task space. Furthermore, we implement a balanced contrastive module to address C1 and C2, which effectively balances the gradient contributions across head classes and tail classes. This module reduces the loss of tail tasks while ensuring the performance of head tasks, thus controlling the range of losses across all tasks.

We systematically evaluate the performance of TAIL2LEARN with eleven baseline models on six real-world datasets for long-tail classification on graphs. The results demonstrate the effectiveness of TAIL2LEARN and verify our theoretical findings.

## 2 PRELIMINARY

In this section, we introduce the background and give the formal problem definition. Table 4 in Appendix A summarizes the main notations used in this paper. We represent a graph as $\mathcal{G} = (\mathcal{V}, \mathcal{E}, \mathbf{X})$, where $\mathcal{V}$ represent the set of nodes, $\mathcal{E} \subseteq \mathcal{V} \times \mathcal{V}$ represent the set of edges, $\mathbf{X} \in \mathbb{R}^{n \times d}$ represent the node feature matrix, $n$ is the number of nodes, and $d$ is the feature dimension. $\mathbf{A} \in \{0, 1\}^{n \times n}$ is the adjacency matrix, where $\mathbf{A}_{ij} = 1$ if there is an edge $\mathbf{e}_{ij} \in \mathcal{E}$ from $\mathbf{v}_i$ to $\mathbf{v}_j$ in $\mathcal{G}$ and $\mathbf{A}_{ij} = 0$ otherwise. $\mathcal{Y} = \{y_1, \ldots, y_n\}$ is the set of labels, $y_i \in \{1, \ldots, T\}$ is the label of the $i^{\text{th}}$ node, and there are $T$ classes in total.

**Long-Tail Classification.** It refers to the classification problem in the presence of a massive number of classes, highly-skewed class-membership distribution, and label scarcity. Here $\mathcal{D} = \{(\mathbf{x}_i, y_i)\}_{i=1}^n$ represents a dataset with long-tail distribution. We define $\mathcal{D}_t$ as the set of instances belonging to class $t$, and $T$ can be notably large. Without the loss of generality, we have $\mathcal{D} = \{\mathcal{D}_1, \mathcal{D}_2, \ldots, \mathcal{D}_T\}$, where $|\mathcal{D}_1| \geq |\mathcal{D}_2| \geq \cdots \gg |\mathcal{D}_T|$, $\sum_{t=1}^T |\mathcal{D}_t| = n$. Tail classes may encounter label scarcity, having few or even only one instance, while head classes have abundant instances. To measure the skewness of

long-tail distribution, Wu et al. (2021) introduces the Class-Imbalance Ratio as $\frac{\min_t(|\mathcal{D}_t|)}{\max_t(|\mathcal{D}_t|)}$, i.e., the ratio of the size of the largest majority class to the size of the smallest minority class.

**Long-Tailedness Ratio.** We consider a graph $\mathcal{G}$ with long-tail distribution. While Class-Imbalance Ratio (Wu et al., 2021) considers the imbalanced data distribution, it overlooks the task complexity in the task of long-tail classification. As the number of categories increases, the difficulty of the classification task therefore increases. For example, we down-sampled 7 categories from the original Cora-Full dataset, as shown in Figure 2. Although the class-imbalance ratio remains the same, i.e., 0.02 for both the original and down-sampled datasets, the task complexity varies significantly, i.e., 70 classes in Figure 2 (a) v.s 7 classes in Figure 2 (b). For this reason, we introduce a novel quantile-based metric named long-tailedness ratio to jointly quantify the class-imbalance ratio and task complexity for the long-tail datasets.

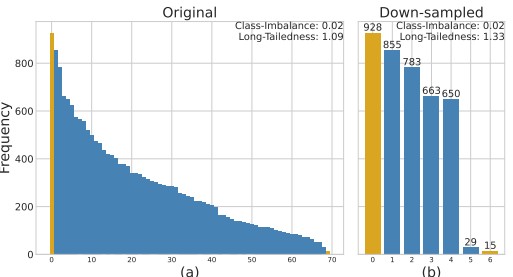

Figure 2: Comparison between two long-tail distribution metrics on (a) the hard case of the original Cora-Full dataset and (b) the easy case of the down-sampled Cora-Full dataset. We observe that the class-imbalance ratio falls short in characterizing the task complexity of two datasets, while the long-tailedness ratio does.

The formal definition of long-tailedness ratio is provided as follows:

**Definition 1** (Long-Tailedness Ratio). *Suppose we have a dataset $\mathcal{D}$ with long-tail categories that follow a descending order in terms of the number of instances. The long-tailedness ratio is*

$$Ratio_{LT}(p) = \frac{Q(p)}{T - Q(p)}. \tag{1}$$

*where $Q(p) = min\{y : Pr(\mathcal{Y} \leq y) = p, 1 \leq y \leq T\}$ is the quantile function of order $p \in (0,1)$ for variable $\mathcal{Y}$, $T$ is the number of categories. The numerator represents the number of categories to which $p$ percent instances belong, and the denominator represents the number of categories to which the else $(1-p)$ percent instances belong in $\mathcal{D}$.*

Essentially, the long-tailedness ratio implies the task complexity of long-tail classification and characterizes two properties of $\mathcal{D}$: (1) class-membership skewness, (2) # of classes. Intuitively, the higher the skewness of data distribution, the lower the ratio will be; the higher the complexity of the tasks (i.e., massive number of classes), the lower the long-tailedness ratio. Figure 2 provides a case study on Cora-Full dataset by comparing long-tailedness ratio and class-imbalance ratio (Wu et al., 2021). In general, we observe that long-tailedness ratio better characterizes the differences on the original Cora dataset ($Ratio_{LT}(0.8) = 1.09$) and its down-sampled dataset ($Ratio_{LT}(0.8) = 1.33$). In our implementation, we choose $p = 0.8$ following the Pareto principle (Pareto et al., 1971). In Appendix B, we additionally offer insights into the utilization of the long-tailedness ratio for enhanced comprehension of long-tail datasets and as a guiding factor for model selection in practice.

## 3 TAIL2LEARN MODEL

### 3.1 THEORETICAL ANALYSIS

In this paper, we consider long-tail problems with data imbalance and massive categories, an area with limited theoretical exploration. For the first time, we propose to reformulate long-tail problems in the manner of multi-task learning, thereby leveraging the theoretical foundation of multi-task learning to gain insights into long-tail problems. In particular, we view the classification for each category as a learning task[2] on graph $\mathcal{G}$. A key assumption of multi-task learning is task relatedness, i.e., relevant tasks should share similar model parameters. Similarly, in long-tail learning, we aim to learn the related tasks (categories) concurrently to potentially enhance the performance of each task (categories). We propose to formulate the hypothesis $g$ of long-tail model as $g = \{f_t\}_{t=1}^{T} \circ h$, where $\circ$ is the functional composition, $g_t(x) = f_t \circ h(x) \equiv f_t(h(x))$ for each classification task.

---

[2]Here we consider the number of tasks to be the number of categories for simplicity, while in Sec. 3.2 the number of tasks can be smaller than the number of categories after the task grouping operation.

The function $h : \mathbf{X} \to \mathbb{R}^K$ is the representation extraction function shared across different tasks, $f : \mathbb{R}^K \to \mathbb{R}$ is the task-specific predictor, and $K$ is the dimension of the hidden layer. The training set for the $t^{\text{th}}$ task $\mathcal{D}_t = \{(\mathbf{x}_i^t, y_i^t)\}_{i=1}^{n_t}$ contains $n_t$ annotated nodes, $\mathbf{x}_i^t$ is the $i^{\text{th}}$ training node in class $t$, and $y_i^t = t$ for all $i$. The task-averaged risk of representation $h$ and predictors $f_1, \ldots, f_T$ is defined as $\epsilon(h, f_1, \ldots, f_T)$, and the corresponding empirical risk is defined as $\hat{\epsilon}(h, f_1, \ldots, f_T)$. To characterize the performance of head and tail categories in our problem setting, we formally define the loss range of $f_1, \ldots, f_T$ in Definition 2:

**Definition 2** (Loss Range). *The loss range of the $T$ predictors $f_1, \ldots, f_T$ is defined as the difference between the lowest and highest values of the loss function across all tasks.*

$$\texttt{Range}(f_1, \ldots, f_T) = \max_t \frac{1}{n_t} \sum_{i=1}^{n_t} l(f_t(h(\mathbf{x}_i^t)), y_i^t) - \min_t \frac{1}{n_t} \sum_{i=1}^{n_t} l(f_t(h(\mathbf{x}_i^t)), y_i^t), \quad (2)$$

*where $l(\cdot, \cdot)$ is a loss function. For node classification task, $l(\cdot, \cdot)$ refers to cross-entropy loss.*

In the scenario of long-tail class-membership distribution, there often exists a tension between maintaining head class performance and improving tail class performance (Zhang et al., 2021). Minimizing the losses of the head classes may lead to a biased model, which increases the losses of the tail classes. Under the premise that the model could keep a good performance on head tasks, we conjecture that controlling the loss range could improve the performance on tail tasks and lead to a better generalization performance of the model. To verify our idea, we drive the loss range-based generalization error bound for long-tail categories on graphs in the following Theorem 1, and the proof is provided in Appendix C.

**Theorem 1** (Generalization Error Bound). *Given the node embedding extraction function $h \in \mathcal{H}$ and the task-specific classifier $f_1, \ldots, f_T \in \mathcal{F}$, with probability at least $1 - \delta, \delta \in [0, 1]$, we have*

$$\mathcal{E} - \hat{\mathcal{E}} \leq \sum_t \left( \frac{c_1 LG(\mathcal{H}(\mathbf{X}))}{n_t} + \frac{c_2 \sup_{h \in \mathcal{H}} \|h(\mathbf{X})\| \texttt{Range}(f_1, \ldots, f_T)}{n_t} + \sqrt{\frac{9 \ln(2/\delta)}{2n_t}} \right), \quad (3)$$

*where $\mathbf{X}$ is the node feature, $T$ is the number of tasks, $n_t$ is the number of nodes in task $t$, $L$ is Lipschitz constant, $G$ is Gaussian complexity, and $c_1$ and $c_2$ are universal constants.*

**Remark:** Theorem 1 implies that the generalization error is upper bounded by the Gaussian complexity of the shared representation extraction $h \in \mathcal{H}$, the loss range of the task-specific predictors $f_1, \ldots, f_T$, the number of total tasks and the sum of their reciprocal. Controlling the complexity of task space and the loss range $\texttt{Range}(f_1, \ldots, f_T)$ can tighten this upper bound, which motivates the development of TAIL2LEARN in the following subsection.

### 3.2 TAIL2LEARN FRAMEWORK

The overview of TAIL2LEARN is presented in Figure 3, which consists of two major modules: M1. hierarchical task grouping and M2. long-tail balanced contrastive learning. Specifically, the theoretical analysis in Theorem 1 inspires that reducing the number of total tasks can improve the generalization ability. Thus, M1 is designed to control the complexity of task space and capture the information shared across tasks by grouping tasks into the hypertasks to improve overall performance. Another key message from Theorem 1 is that reducing the loss of tail tasks while ensuring the performance of head tasks controls the range of losses across all tasks and thus improves the generalization ability. Therefore, in M2, we designed a long-tail balanced contrastive loss to balance the head classes and the tail classes. We also provide the optimization and pseudo-code of TAIL2LEARN in Appendix D.

**M1. Hierarchical Task Grouping.** We propose to address C2 (Label scarcity) and C3 (Task complexity) for characterizing long-tail categories on graphs by leveraging the information learned in one category to help train another category. Inspired by multi-task learning, we implement task grouping (Song et al., 2022b) to share information across different tasks via hierarchical pooling (Ying et al., 2018; Gao & Ji, 2019). The core idea of hierarchical pooling is to choose the important nodes and preserve the original connections between chosen nodes as edges to generate a coarsened graph. As shown in Figure 4, the task grouping operation is composed of two steps: (Step 1) we group nodes into several tasks, and (Step 2) learn the embeddings of the task prototypes. This operation can be easily generalized to the $l^{\text{th}}$ layers, which leads to the hierarchical task grouping.

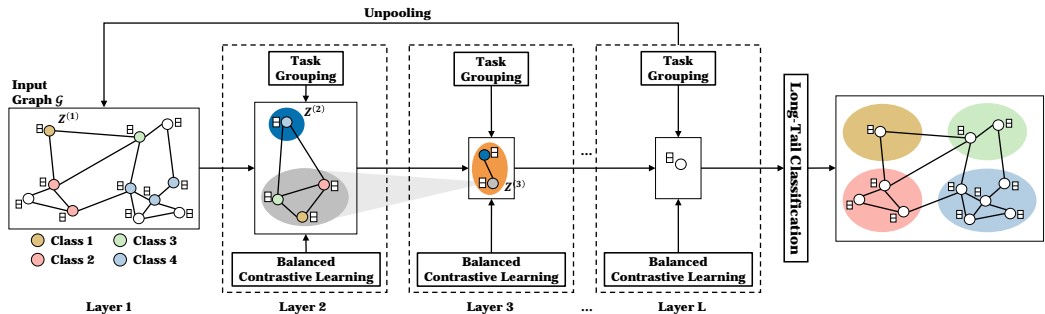

Figure 3: The proposed TAIL2LEARN framework with $L$ task-grouping layers.

Specifically, we first generate a low-dimensional node embedding vector for each node $\mathbf{Z}^{(1)} = (\mathbf{z}_1^{(1)}, \ldots, \mathbf{z}_n^{(1)})$ via graph convolutional network (GCN) (Kipf & Welling, 2016) layers. Next, we group nodes into tasks (with the same number of categories), and then group these tasks into hypertasks by stacking several task grouping layers. The $l^{\text{th}}$ task grouping layer is defined as:

$$\mathcal{I} = \text{TOP-RANK}(\text{PROJ}(\mathbf{Z}^{(l)}), T^{(l)}),$$

$$\mathbf{X}^{(l+1)} = \mathbf{Z}^{(l)}(\mathcal{I}, :) \odot \left( \text{PROJ}(\mathbf{Z}^{(l)}) \mathbf{1}_d^T \right), \quad (4)$$

$$\mathbf{A}^{(l+1)} = \mathbf{A}^{(l)}(\mathcal{I}, \mathcal{I}),$$

where $l = 1, \ldots L$ is the layer of hierarchical task grouping. We generate a new graph with selected important nodes, where these nodes serve as the prototypes of tasks (hypertasks), and $\mathcal{I}$ is the indexes of the selected nodes. $\text{PROJ}(\cdot, \cdot)$ is a projection function to score the node importance by mapping each embedding $\mathbf{z}_i^{(l)}$ to a scalar. $\text{TOP-RANK}$ identifies top $T^{(l)}$ nodes with the highest value after projection.

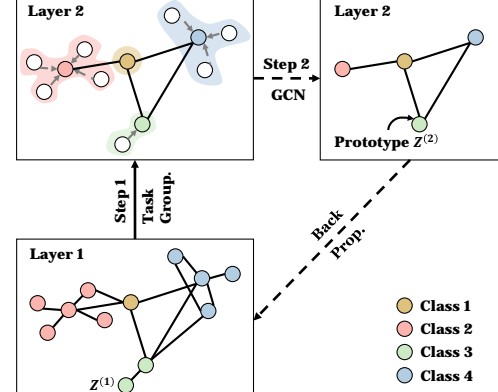

Figure 4: An illustrative figure for M1 with two task-grouping layers. Step 1: nodes are first grouped into four tasks (each representing a class). Step 2: We learn the embeddings of the task prototypes. Finally, the node embeddings are updated by back-propagation.

The connectivity between the selected nodes remains as edges of the new graph, and the new adjacency matrix $\mathbf{A}^{(l+1)}$ and feature matrix $\mathbf{X}^{(l+1)}$ are constructed by row and/or column extraction. The subsequent GCN layer outputs the embeddings $\mathbf{Z}^{(l+1)}$ of the new graph based on $\mathbf{X}^{(l+1)}$ and $\mathbf{A}^{(l+1)}$. Notably, $\mathbf{Z}^{(1)}$ is the node embeddings, $\mathbf{Z}^{(2)}$ is the embeddings of the task prototypes corresponding to the categories, and $\mathbf{Z}^{(l)}(l > 2)$ is the hypertask prototype embeddings.

The number of tasks $T^{(l)}$ represents the level of abstraction of task grouping, and decreases as the task grouping layer gets deeper. In high level layers ($l > 1$), the number of tasks may be smaller than the number of categories. By controlling $T^{(l)}$, information shared across tasks can be obtained to alleviate the *task complexity*, which is associated with characterizing an increasing number of categories. Meanwhile, nodes that come from different categories with high-level semantic similarities can be assigned to one task. By sharing label information with other different categories within the same hypertask, the problem of *label scarcity* can be alleviated. In layer 2 (Figure 4), we consider a special case of 2 head classes (i.e., class 2 and 4) and 2 tail classes (i.e., class 1 and 3). By grouping the prototypes of class 1, 2, and 3 into the same hypertask at a later task grouping layer, our method will automatically assign a unique hypertask label to all nodes belonging to the three classes.

In order to well capture the hierarchical structure of tasks and propagate information across different tasks, we need to restore the original resolutions of the graph to perform node classification. Specifically, we stack the same number of unpooling layers as the task grouping layers, which up-samples the features to restore the original resolutions of the graph.

$$\mathbf{X}^{(l+1)} = \text{DIST} \left( 0_{n \times d}, \mathbf{X}^{(l+1)}, \mathcal{I} \right), \quad (5)$$

where $\text{DIST}$ restore the selected graph to the resolution of the original graph by distributing row vectors in $X^{(l+1)}$ into matrix $0_{n \times d}$ based on the indices $\mathcal{I}$, where $0_{n \times d}$ represents the initially all-zeros

feature matrix, $X^{(l+1)} \in \mathbb{R}^{T^{(l)} \times d}$ represents the feature matrix of the current graph, $\mathcal{I}$ represents the indices of selected nodes in the corresponding task grouping layer. Finally, the corresponding blocks of the task grouping and unpooling layers are skip-connected by feature addition, and the final node embeddings are passed to an MLP layer for final predictions.

**M2. Long-Tail Balanced Contrastive Learning.** To better handle long-tail classification and solve C1 (High-skewed data distribution) and C2 (Label scarcity), we customize a principled graph contrastive learning strategy for M1 (Hierarchical task grouping) by passing labels across layers. Graph contrastive learning (GCL) (Xu et al., 2021; Hassani & Khasahmadi, 2020; Qiu et al., 2020; Zhu et al., 2021) has achieved supreme performance in various tasks with graph-structure data. It aims to learn efficient graph/node representations by constructing positive and negative instance pairs. In this paper, we propose to incorporate supervision signals into each layer of graph contrastive learning. Specifically, we employ supervised contrastive loss $\mathcal{L}_{SCL}$ on the labeled node to augment the original graph. It allows joint consideration of head and tail categories and balancing their contributions, alleviates the challenge of *high-skewed data distribution*. Additionally, we employ balanced contrastive loss $\mathcal{L}_{BCL}$ on each layer of TAIL2LEARN. We group all nodes on the graph into several tasks, facilitating label information to be passed among similar nodes during task grouping. These tasks are subsequently grouped into higher-level hypertasks, enabling label sharing across layers. Through the sharing of label information across nodes and layers, we effectively mitigate the challenge of *label scarcity* in tail categories.

Next, we introduce supervised contrastive loss $\mathcal{L}_{SCL}$ on the restored original graph. It makes node pairs of the same category close to each other while pairs not belonging to the same category far apart. The mathematical form of the loss function $\mathcal{L}_{SCL}$ on $i^{\text{th}}$ node $\mathbf{z}_i$ can be expressed as follows:

$$\mathcal{L}_{SCL}(\mathbf{z}_i) = -\frac{1}{n_t - 1} \times \sum_{j \in \mathcal{V}_t \setminus i} \log \frac{\exp(\mathbf{z}_i \cdot \mathbf{z}_j / \tau)}{\sum_{1 \leq q \leq T} \frac{1}{n_q} \sum_{k \in \mathcal{V}_q} \exp(\mathbf{z}_i \cdot \mathbf{z}_k / \tau)}, \tag{6}$$

where $\mathbf{z}_i$ belongs to class $t$, $\mathcal{V}_t$ is all the nodes belonging to class $t$, $z_k$ represents the embedding of the $k^{\text{th}}$ node, and temperature $\tau$ controls the strength of penalties on negative node. $\mathcal{L}_{SCL}$ reduces the proportion of contributions from head classes and highlights the importance of tail classes to alleviate the bias caused by high-skewed data distribution.

Moreover, we introduce balanced contrastive loss $\mathcal{L}_{BCL}$ on a coarsened graph, where each node represents a task prototype. For $l^{\text{th}}$ task grouping layer, we group tasks in layer $l$ into $T^{(l)}$ hypertasks and calculate the balanced contrastive loss based on the task embeddings $\mathbf{Z}^{(l)}$ and the hypertask prototypes $\mathbf{Z}^{(l+1)}$. It pulls the task embeddings together with their corresponding hypertask prototypes and pushes them away from other prototypes. $\mathcal{L}_{BCL}$ on $i^{\text{th}}$ node $\mathbf{z}_i$ can be expressed as follows[3]:

$$\mathcal{L}_{BCL}(\mathbf{z}_i) = -\frac{1}{n_t} \times \sum_{j \in \mathcal{V}_t \setminus i} \log \frac{\exp(\mathbf{z}_i \cdot \mathbf{z}_j / \tau)}{\sum_{1 \leq q \leq T} \frac{1}{n_q + 1} \sum_{k \in \mathcal{V}_q} \exp(\mathbf{z}_i \cdot \mathbf{z}_k / \tau)}, \tag{7}$$

where we suppose $\mathbf{z}_i$ belongs to hypertask $t$, here $\mathcal{V}_t$ are all the nodes within the $t^{\text{th}}$ hypertask including the hypertask prototype $\mathbf{z}_t^{(l+1)}$, $n_t$ represents the number of nodes in hypertask $t$, $\mathbf{z}_k = \mathbf{z}_k^{(l)}$ represents the embedding of the $k^{\text{th}}$ node, and $\tau$ is the temperature. Therefore, $\mathcal{L}_{BCL}$ solves the long-tail classification in two aspects: (1) It potentially controls the range of losses for different tasks. The $n_j + 1$ term in the denominator averages over the nodes of each task in order that each task has an approximate contribution for optimizing; (2) The set of $T$ hypertask prototypes is added to obtain a more stable optimization for balanced contrastive learning.

In summary, M2 combines supervised contrastive loss and balanced contrastive loss. With M2, we alleviate the label scarcity by passing label information across all nodes and all layers; and solve the data imbalance by balancing the performance of the head and tail classes.

Theorem 1 shows that the generalization error of long-tail on graphs can be improved by (1) reducing the loss range across all tasks $\text{Range}(f_1, \ldots, f_T)$, as well as (2) controlling the tasks complexity. Below we give a corollary (proof in Appendix C) to theoretically explain how the two modules in TAIL2LEARN work. The left-hand side of the inequality is the error bound of TAIL2LEARN with $l$ layers, while the right-hand side is the error bounds without using the two modules in Theorem 1.

---

[3]We use the same contrastive loss for each layer. For clarify, we omit layer $(l)$.

**Corollary 1** (Effectiveness of TAIL2LEARN). *After the $l^{th}$ hierarchical task grouping layer, we group nodes into $T^{(l)}$ tasks with $T^{(l)} < T$, where $T$ is the number of classes. In addition, we can learn the task-specific predictors $f_1^{(l)}, \ldots, f_T^{(l)}$ with $Range(f_1^{(l)}, \ldots, f_T^{(l)}) < Range(f_1, \ldots, f_T)$ by long-tail balanced contrastive learning. Then we have that the upper bound of the error for TAIL2LEARN with $l$ layers is smaller than the upper bound without the two modules, i.e.,*

$$
\sum_t \left( \frac{c_1 LG(\mathcal{H}(\mathbf{X}))}{n_t^{(l)}} + \frac{c_2 \sup_{h \in \mathcal{H}} \|h(\mathbf{X})\| Range(f_1^{(l)}, \ldots, f_T^{(l)})}{n_t^{(l)}} + \sqrt{\frac{9 \ln(2/\delta)}{2 n_t^{(l)}}} \right)
$$
$$
\leq \sum_t \left( \frac{c_1 LG(\mathcal{H}(\mathbf{X}))}{n_t} + \frac{c_2 \sup_{h \in \mathcal{H}} \|h(\mathbf{X})\| Range(f_1, \ldots, f_T)}{n_t} + \sqrt{\frac{9 \ln(2/\delta)}{2 n_t}} \right),
\tag{8}
$$

*where $h \in \mathcal{H}$ is the shared GCN layer for representation learning. The number of instances in the $t^{th}$ class for layer $l$ is $n_t^{(l)}$, we have $\sum_t n_t^{(l)} = \sum_t n_t = n$, where $n$ is the total number of instances.*

**Remark:** Corollary 1 theoretically demonstrate the effectiveness of TAIL2LEARN. Our algorithm leads to a significantly improved error bound in long-tail classification on graphs, by controlling the complexity of task space in M1 and controlling the loss range $Range(f_1, \ldots, f_T)$ in M2.

## 4 EXPERIMENTS

We evaluate the effectiveness of TAIL2LEARN on six benchmark datasets, and it exhibits superior performances compared to various state-of-the-art baselines (Section 4.2). We demonstrate the necessity of each component of TAIL2LEARN in ablation studies (Section 4.3). We also report the parameter and complexity sensitivity of TAIL2LEARN which shows that TAIL2LEARN achieves a convincing performance with minimal tuning efforts and is scalable (Appendix F).

### 4.1 EXPERIMENT SETUP

**Datasets:** We evaluate our proposed framework on Cora-Full (Bojchevski & Günnemann, 2018), BlogCatalog (Tang & Liu, 2009), Email (Yin et al., 2017), Wiki (Mernyei & Cangea, 2020), Amazon-Clothing (McAuley et al., 2015), and Amazon-Electronics (McAuley et al., 2015) datasets to perform node classification task. The first four datasets naturally have smaller $\mathtt{Ratio}_{LT}$, indicating higher long-tail; while the last two datasets have larger $\mathtt{Ratio}_{LT}$, requiring the manual process to make them harsh long-tail with $\mathtt{Ratio}_{LT} \approx 0.25$. Our proposed $\mathtt{Ratio}_{LT}$ reflects a similar trend compared to the class-imbalance ratio, but offers a more accurate measurement by considering the total number of categories. Further details and statistics of the six datasets are in Appendix E.1.

**Comparison Baselines:** We compare TAIL2LEARN with five imbalanced classification methods and six GNN-based long-tail classification methods. The details of baselines are in Appendix E.2.

- Imbalanced classification methods: Origin (i.e., GCN (Kipf & Welling, 2017)), Over-sampling (Chawla, 2003), Re-weighting (Yuan & Ma, 2012), SMOTE (Chawla et al., 2002), and Embed-SMOTE (Ando & Huang, 2017).
- GNN-based long-tail classification methods: Two popular variants of GraphSMOTE (Zhao et al., 2021) (GraphSMOTE$_T$ and GraphSMOTE$_O$), GraphMixup (Wu et al., 2021), ImGAGN (Qu et al., 2021), GraphENS (Park et al., 2022), and LTE4G (Yun et al., 2022).

**Implementation Details:** We run all the experiments with 10 random seeds and report the evaluation metrics along with standard deviations. Considering the long-tail class-membership distribution, balanced accuracy (bAcc), Macro-F1 and Geometric Means (G-Means) are used as the evaluation metrics, and accuracy (Acc) as the traditional metric. Parameter settings are in Appendix E.3.

### 4.2 PERFORMANCE ANALYSIS

**Overall Evaluation.** We compared TAIL2LEARN with eleven methods on six real-world graphs, and the performance of node classification is reported in Table 1 and Table 2. In general, we have the following observations: (1) TAIL2LEARN consistently performs well on all datasets under various long-tail settings, and especially outperforms other baselines on harsh long-tail settings (e.g., $\mathtt{Ratio}_{LT}(0.8) \approx 0.25$), which demonstrates the effectiveness and generalizability of our model. More precisely, taking the Amazon-Electronics dataset (167 categories and follows the Pareto distribution with "80-20 Rule") as an example, the improvement of our model on bAcc (Acc) is 12.9%

| Method | Cora-Full | | | | BlogCatalog | | | |
|---|---|---|---|---|---|---|---|---|
| | bAcc | Macro-F1 | G-Means | Acc | bAcc | Macro-F1 | G-Means | Acc |
| *Classical* — Origin | 52.8 ± 0.6 | 54.5 ± 0.7 | 72.5 ± 0.4 | 62.7 ± 0.5 | 7.1 ± 0.4 | 7.3 ± 0.4 | 26.4 ± 0.7 | 15.1 ± 1.0 |
| Over-sampling | 52.7 ± 0.7 | 54.4 ± 0.6 | 72.4 ± 0.5 | 62.7 ± 0.4 | 7.1 ± 0.3 | 7.2 ± 0.3 | 26.3 ± 0.6 | 15.1 ± 1.2 |
| Re-weight | 52.9 ± 0.5 | 54.4 ± 0.5 | 72.5 ± 0.3 | 62.6 ± 0.4 | 7.2 ± 0.4 | 7.3 ± 0.5 | 26.4 ± 0.8 | 15.1 ± 0.8 |
| SMOTE | 52.7 ± 0.6 | 54.4 ± 0.5 | 72.4 ± 0.4 | 62.7 ± 0.4 | 7.1 ± 0.4 | 7.2 ± 0.5 | 26.3 ± 0.8 | 15.3 ± 1.2 |
| Embed-SMOTE | 52.9 ± 0.5 | 54.4 ± 0.5 | 73.9 ± 0.4 | 62.6 ± 0.4 | 7.1 ± 0.5 | 7.3 ± 0.5 | 26.3 ± 0.9 | 14.8 ± 0.8 |
| *GNN-based* — $\text{GraphSMOTE}_T$ | 54.2 ± 0.8 | 54.7 ± 0.8 | 73.4 ± 0.6 | 62.1 ± 0.6 | 8.6 ± 0.4 | 8.5 ± 0.5 | 28.9 ± 0.7 | 18.3 ± 1.1 |
| $\text{GraphSMOTE}_O$ | 54.1 ± 0.8 | 54.5 ± 0.7 | 73.3 ± 0.5 | 62.0 ± 0.6 | 8.6 ± 0.4 | 8.5 ± 0.4 | 28.9 ± 0.6 | 18.3 ± 0.9 |
| GraphMixup | 53.9 ± 1.3 | 53.9 ± 1.3 | 73.2 ± 0.9 | 61.4 ± 1.2 | 8.0 ± 0.6 | 7.9 ± 0.8 | 27.9 ± 1.2 | 18.8 ± 0.8 |
| ImGAGN | 9.3 ± 1.1 | 6.6 ± 1.0 | 30.2 ± 1.9 | 20.9 ± 2.1 | 6.2 ± 0.6 | 4.9 ± 0.5 | 24.6 ± 1.3 | 20.5 ± 1.3 |
| GraphENS | 55.0 ± 0.6 | 54.2 ± 0.5 | 73.9 ± 0.4 | 62.1 ± 0.4 | 9.0 ± 0.6 | 8.9 ± 0.5 | 30.8 ± 0.9 | 12.8 ± 1.1 |
| LTE4G | 55.8 ± 0.6 | 54.5 ± 0.4 | 74.5 ± 0.4 | 61.6 ± 0.4 | 6.9 ± 0.5 | 6.7 ± 0.6 | 26.0 ± 0.9 | 11.7 ± 1.3 |
| Ours | **55.8 ± 0.5** | **57.1 ± 0.5** | **74.5 ± 0.3** | **64.7 ± 0.7** | **9.8 ± 0.2** | **9.6 ± 0.1** | **30.9 ± 0.4** | **23.2 ± 0.6** |

| Method | Email | | | | Wiki | | | |
|---|---|---|---|---|---|---|---|---|
| | bAcc | Macro-F1 | G-Means | Acc | bAcc | Macro-F1 | G-Means | Acc |
| *Classical* — Origin | 48.9 ± 4.5 | 45.2 ± 4.3 | 69.5 ± 3.2 | **66.7 ± 2.1** | 48.2 ± 1.5 | 49.9 ± 1.9 | 68.6 ± 1.1 | 64.2 ± 0.9 |
| Over-sampling | 48.4 ± 4.2 | 45.4 ± 3.7 | 69.2 ± 3.1 | 66.4 ± 2.0 | 47.3 ± 2.1 | 48.7 ± 2.2 | 67.9 ± 1.5 | 63.6 ± 1.4 |
| Re-weight | 47.9 ± 4.6 | 44.2 ± 4.2 | 68.8 ± 3.4 | 66.3 ± 1.7 | 48.1 ± 2.1 | 49.7 ± 2.5 | 68.5 ± 1.6 | 64.0 ± 1.4 |
| SMOTE | 48.4 ± 4.2 | 45.4 ± 3.7 | 69.2 ± 3.1 | 66.4 ± 2.0 | 47.3 ± 2.1 | 48.7 ± 2.2 | 67.9 ± 1.5 | 63.6 ± 1.4 |
| Embed-SMOTE | 47.9 ± 4.6 | 44.2 ± 4.2 | 68.8 ± 3.3 | 66.2 ± 1.7 | 48.1 ± 2.1 | 49.7 ± 2.5 | 68.5 ± 1.6 | 63.9 ± 1.4 |
| *GNN-based* — $\text{GraphSMOTE}_T$ | 43.4 ± 2.9 | 39.1 ± 2.8 | 65.5 ± 2.2 | 60.4 ± 1.5 | 50.3 ± 1.7 | 51.8 ± 2.2 | 70.1 ± 1.2 | 65.8 ± 0.9 |
| $\text{GraphSMOTE}_O$ | 42.3 ± 3.1 | 38.3 ± 2.9 | 64.7 ± 2.4 | 60.1 ± 2.3 | 49.6 ± 2.3 | 51.1 ± 2.7 | 69.6 ± 1.7 | 65.5 ± 1.2 |
| GraphMixup | 43.2 ± 2.3 | 38.1 ± 2.3 | 65.4 ± 1.7 | 60.1 ± 1.7 | 50.3 ± 2.9 | 51.2 ± 2.9 | 70.0 ± 2.1 | 65.1 ± 1.3 |
| ImGAGN | 27.6 ± 3.4 | 26.8 ± 2.9 | 52.0 ± 3.2 | 46.5 ± 3.5 | 41.2 ± 5.7 | 42.3 ± 6.4 | 63.2 ± 4.9 | 65.5 ± 5.8 |
| GraphENS | 50.5 ± 3.1 | 43.7 ± 3.3 | **71.1 ± 2.2** | 62.0 ± 2.7 | 50.8 ± 3.3 | 50.1 ± 3.4 | 70.3 ± 2.4 | 61.7 ± 4.4 |
| LTE4G | 46.4 ± 2.5 | 39.3 ± 2.4 | 67.8 ± 1.8 | 57.8 ± 3.1 | 51.0 ± 2.9 | 49.7 ± 1.9 | 70.5 ± 2.1 | 60.4 ± 2.1 |
| Ours | **50.5 ± 3.0** | **46.6 ± 3.0** | 70.7 ± 2.1 | 65.4 ± 1.7 | **52.8 ± 2.0** | **54.1 ± 2.3** | **71.9 ± 1.4** | **67.2 ± 1.1** |

Table 1: Comparison of different methods in node classification task on natural datasets.

| Method | Amazon-Clothing | | | | Amazon-Electronics | | | |
|---|---|---|---|---|---|---|---|---|
| | bAcc | Macro-F1 | G-Means | Acc | bAcc | Macro-F1 | G-Means | Acc |
| *Classical* — Origin | 9.9 ± 0.2 | 9.5 ± 0.2 | 31.3 ± 0.3 | 9.9 ± 0.2 | 16.9 ± 0.2 | 15.2 ± 0.2 | 41.0 ± 0.3 | 16.9 ± 0.2 |
| Over-sampling | 9.9 ± 0.2 | 9.5 ± 0.2 | 31.3 ± 0.3 | 9.9 ± 0.2 | 16.8 ± 0.1 | 15.1 ± 0.1 | 40.9 ± 0.2 | 16.8 ± 0.1 |
| Re-weight | 10.0 ± 0.2 | 9.6 ± 0.2 | 31.4 ± 0.3 | 10.0 ± 0.2 | 17.0 ± 0.2 | 15.2 ± 0.2 | 41.1 ± 0.3 | 17.0 ± 0.2 |
| SMOTE | 10.0 ± 0.1 | 9.5 ± 0.2 | 31.4 ± 0.2 | 10.0 ± 0.1 | 16.9 ± 0.2 | 15.1 ± 0.2 | 41.0 ± 0.3 | 16.9 ± 0.2 |
| Embed-SMOTE | 9.9 ± 0.2 | 9.5 ± 0.2 | 31.3 ± 0.3 | 9.9 ± 0.2 | 17.0 ± 0.2 | 15.2 ± 0.2 | 41.1 ± 0.3 | 17.0 ± 0.2 |
| *GNN-based* — $\text{GraphSMOTE}_T$ | 11.7 ± 0.2 | 10.4 ± 0.3 | 34.0 ± 0.3 | 11.7 ± 0.2 | 18.2 ± 0.2 | 15.6 ± 0.2 | 42.5 ± 0.2 | 18.2 ± 0.2 |
| $\text{GraphSMOTE}_O$ | 11.7 ± 0.2 | 10.4 ± 0.3 | 34.0 ± 0.3 | 11.7 ± 0.2 | 18.2 ± 0.2 | 15.5 ± 0.2 | 42.5 ± 0.2 | 18.2 ± 0.2 |
| GraphMixup | 10.9 ± 0.5 | 9.3 ± 0.7 | 32.8 ± 0.7 | 10.9 ± 0.5 | 18.1 ± 0.4 | 15.5 ± 0.5 | 42.5 ± 0.5 | 18.1 ± 0.4 |
| ImGAGN | 12.9 ± 0.2 | 9.2 ± 0.1 | 35.7 ± 0.2 | 12.9 ± 0.2 | 13.7 ± 0.2 | 11.0 ± 0.0 | 36.9 ± 0.2 | 13.7 ± 0.2 |
| GraphENS | 11.6 ± 2.7 | 10.9 ± 2.7 | 33.6 ± 4.3 | 11.6 ± 2.7 | 19.2 ± 3.8 | 17.2 ± 3.6 | 43.5 ± 4.4 | 19.2 ± 3.8 |
| LTE4G | 15.5 ± 0.3 | 16.0 ± 0.5 | 39.1 ± 0.3 | 15.5 ± 0.3 | 20.9 ± 0.3 | 19.9 ± 0.3 | 45.7 ± 0.3 | 20.9 ± 0.3 |
| Ours | **17.1 ± 0.5** | **16.8 ± 0.6** | **41.1 ± 0.6** | **17.1 ± 0.5** | **23.6 ± 0.9** | **21.0 ± 1.3** | **48.5 ± 1.0** | **23.6 ± 0.9** |

Table 2: Comparison of different methods in node classification task on semi-synthetic long-tail datasets with long-tailedness ratio $\texttt{Ratio}_{LT}(0.8) \approx 0.25$.

compared to the second best model (LTE4G). Its implies that TAIL2LEARN can not only solve the highly skewed data but also capture massive number of classes. (2) Classical long-tail learning methods have the worst performance because they ignore graph structure information and only conduct oversampling or reweighting in the feature space. TAIL2LEARN improves bAcc up to 36.1% on the natural dataset (BlogCatalog) and 71.0% on the manually processed dataset (Amazon-Clothing) compared to the classical long-tail learning methods. (3) GNN-based long-tail learning methods achieved the second best performance (excluding the Email dataset), which implies that it is beneficial to capture or transfer knowledge on the graph topology, but these models ignore massive number of categories. In particular, since ImGAGN only considers the high-skewed distribution, as the number of categories increases (from Wiki to Cora-Full), the model becomes less effective. Our model outperforms these GNN-based methods on almost all the natural datasets and metrics (excluding Email), such as up to 12.9% on the manually processed dataset (Amazon-Electronics).

**Performance on Each Category.** To observe the performance of our model for the long-tail classification, in Figure 1, we plot the model performance (bAcc) on each category. We find that TAIL2LEARN outperforms the original GCN method (fails to consider the long-tail class-membership distribution), especially on the tail classes.

## 4.3 ABLATION STUDY

Table 3 presents the node classification performance on Cora-Full when considering (a) complete TAIL2LEARN (b) hierarchical long-tail category grouping and node classification loss; and (c) only node classification loss. From the results, we have

| Components | | | Cora-Full | | | |
|---|---|---|---|---|---|---|
| M1 | M2 | $\mathcal{L}_{CE}$ | bAcc | Macro-F1 | G-Means | Acc |
| ✓ | ✓ | ✓ | **55.8** ± 0.5 | **57.1** ± 0.5 | **74.5** ± 0.3 | **64.7** ± 0.7 |
| ✓ | | ✓ | 54.5 ± 0.5 | 56.2 ± 0.4 | 73.6 ± 0.3 | 64.5 ± 0.4 |
| | | ✓ | 52.8 ± 0.6 | 54.5 ± 0.7 | 72.5 ± 0.4 | 62.7 ± 0.5 |

Table 3: Ablation study on each component of TAIL2LEARN.

several interesting observations: (1) Long-tail balanced contrastive learning module (M2) leads to an increase in bAcc by 1.9%, which shows its strength in improving long-tail classification by ensuring accurate node embeddings ((a) > (b)). (2) Hierarchical task grouping (M1) helps the model better share information across tasks, it achieves impressive improvement on Cora-Full by up to 3.2% ((b) > (c)). Overall, the ablation study firmly attests both two modules are essential in successful long-tail classification on graphs.

## 5 RELATED WORK

**Long-tail problems.** Long-tail data distributions are common in real-world applications (Zhang et al., 2021). Several methods are proposed to solve the long-tail problem, such as data augmentation methods (Chawla, 2003; Liu et al., 2008) and cost-sensitive methods (Elkan, 2001; Zhou & Liu, 2005; Yuan & Ma, 2012). However, the vast majority of previous efforts focus on independent and identically distributed (i.i.d.) data, which cannot be directly applied to graph data. Recently, several related works for long-tail classification on graphs (Park et al., 2022; Yun et al., 2022; Wu et al., 2021; Shi et al., 2020; Qu et al., 2021; Liu et al., 2021; 2020; Zhang et al., 2022b; Zheng et al., 2022; Zeng et al., 2023; Shi et al., 2021; Li et al., 2022; Qian et al., 2022; Zhang et al., 2022a; Song et al., 2022a) have attracted attention. The first work named GraphSMOTE (Zhao et al., 2021) interpolates tail node embeddings and generates edges utilizing an edge generator. However, the long-tail approaches often lack a theoretical basis. The most relevant work lies in imbalanced classification. Cao et al. (2019) and Kini et al. (2021) presented model-related bounds on the error and the SVM margins, while Yang & Xu (2020) provided the error bound of a linear classifier on data distribution and dimension. In addition, previous long-tail work is experimented under class imbalance settings where the number of classes can be small and the number of minority nodes may not be small; but for long-tail learning, the number of classes is large and the tail nodes are scarce. In this paper, we provide a theoretical analysis of the long-tail problem and conduct experiments on long-tail datasets.

**Graph Neural Networks.** Graph neural networks emerge as state-of-the-art methods for graph representation learning, which capture the structure of graphs. Recently, several attempts have been focused on extending pooling operations to graphs. In order to achieve an overview of the graph structure, hierarchical pooling (Ma et al., 2019; Ranjan et al., 2020; Lee et al., 2019; Ying et al., 2018; Gao & Ji, 2019) techniques attempt to gradually group nodes into clusters and coarsen the graph recursively. Graph U-Nets (Gao & Ji, 2019) proposes a encoder-decoder architecture based on gPool and gUnpool layers. However, these approaches are generally designed to enhance the representation of the whole graph. In this paper, we aim to explore node classification with the long-tail class-membership distribution via hierarchical pooling methods.

## 6 CONCLUSION

In this paper, we investigate long-tail classification on graphs, which intends to improve the performance on both head and tail classes. By formulating this problem in the fashion of multi-task learning, we propose the first generalization bound dominated by the range of losses across all tasks and the total number of tasks. Building upon the theoretical findings, we also present TAIL2LEARN. It is a generic framework with two major modules: M1. Hierarchical task grouping to control the complexity of task space and address C2 (Label scarcity) and C3 (Task complexity); and M2. Long-tail balanced contrastive learning to control the range of losses across all tasks and solve C1 (High-skewed data distribution) and C2 (Label scarcity). Extensive experiments on six real-world datasets, where TAIL2LEARN consistently outperforms state-of-art baselines, demonstrate the efficacy of our model for capturing long-tail categories on graphs. Our code and data are released at `https://anonymous.4open.science/r/Tail2Learn-CE08/`.

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

## A  SYMBOLS AND NOTATIONS

Here we give the main symbols and notations in this paper.

Table 4: Symbols and notations.

| Symbol | Description |
|---|---|
| $\mathcal{G}$ | input graph. |
| $\mathcal{V}$ | the set of nodes in $\mathcal{G}$. |
| $\mathcal{E}$ | the set of edges in $\mathcal{G}$. |
| $\mathbf{X}$ | the node feature matrix of $\mathcal{G}$. |
| $\mathbf{Z}$ | the node embeddings in $\mathcal{G}$. |
| $\mathbf{A}$ | the adjacency matrix in $\mathcal{G}$. |
| $\mathcal{Y}$ | the set of labels in $\mathcal{G}$. |
| $n$ | the number of nodes $|\mathcal{V}|$. |
| $T$ | the number of categories of nodes $\mathcal{V}$. |
| $\texttt{Ratio}_{LT}$ | the long-tailedness ratio. |

## B  DETAILS OF $\texttt{RATIO}_{LT}(p)$

To better characterize class-membership skewness and number of classes, we introduce a novel quantile-based metric named long-tailedness ratio for the long-tail datasets.

$$\texttt{Ratio}_{LT}(p) = \frac{Q(p)}{T - Q(p)}$$

where $Q(p) = min\{y : Pr(\mathcal{Y} \leq y) = p, 1 \leq y \leq T\}$ is the quantile function of order $p \in (0, 1)$ for variable $\mathcal{Y}$, $T$ is the number of categories. The numerator represents the number of categories to which $p$ percent instances belong, and the denominator represents the number of categories to which the else $(1 - p)$ percent instances belong in $\mathcal{D}$.

The hyperparameter $p$ allows end users to control the number of classes in the head of the long-tail distribution. If there is no specific definition of the head class in certain domains, we suggest simply following the Pareto principle ($p = 0.8$). Using the same $p$ value for two long-tail datasets allows us to compare the complexity. Otherwise, if the $\texttt{Ratio}_{LT}(p)$ of two datasets are measured based on different $p$ values, they are not comparable. If there is a specific definition of the head class in certain domains, we can directly calculate the number of head classes and thus infer the $p$ value.

In addition, in light of class-imbalance ratio and long-tailedness ratio, we gain a better understanding of the datasets and methods to use. (1) High class-imbalance ratio and low $\texttt{Ratio}_{LT}$ imply high-skewed data distribution, and we may encounter a large number of categories. In such situations, a long-tail method that is designed for data imbalance and an extreme number of classes may be necessary to achieve optimal results. (2) High class-imbalance ratio and high $\texttt{Ratio}_{LT}$ suggest that the task complexity is low with a relatively small number of categories and the dataset may be imbalanced. Therefore, imbalanced classification approaches such as re-sampling or re-weighting may be effective. (3) Low class-imbalance ratio and low $\texttt{Ratio}_{LT}$ imply high task complexity but relatively balanced samples. In such cases, extreme classification methods would be preferred. (4) Low class-imbalance ratio and high $\texttt{Ratio}_{LT}$ suggest that the dataset may not follow a long-tail distribution, and ordinary machine learning methods may achieve great performance.

## C  DETAILS OF THEORETICAL ANALYSIS

We obtain the range-based generalization error bound for long-tail categories in the following three steps: (S1) giving the loss-related generalization error bound based on the Gaussian complexity-based bound in Lemma 1; (S2) giving the hypothesis $g$-related generalization error bound based on the loss-related error bound in S1 and the property of Gaussian complexity in Lemma 2; (S3) deriving the generalization error bound related to representation extraction $h$ and the range of task-specific predictors $f_1, \ldots, f_T$ (Theorem 1) based on the obtained hypothesis $g$-related bound in S2 and the chain rule of gaussian complexity in Lemma 3.

Based on Maurer et al. (2016), we can derive the Gaussian complexity-based bound on the training set $\mathbf{X}$ (S1) as follows.

**Lemma 1** (Gaussian Complexity-Based Bound). *Let $\mathcal{F}$ be a class of functions $f : \mathbf{X} \to [0,1]^T$, and $\mathbf{x}_i^t$ represents $i^{th}$ instances belonging to class $t$. Then, with probability greater than $1 - \delta$ and for all $f \in \mathcal{F}$, we have the following bound*

$$\frac{1}{T} \sum_t \left( \mathbb{E}_{\mathbf{X} \sim \mu_t} [f_t(\mathbf{X})] - \sum_i \frac{1}{n_t} f_t \left( \mathbf{x}_i^t \right) \right) \leq \sum_t \left( \frac{\sqrt{2\pi} G(\mathcal{Y})}{n_t} + \sqrt{\frac{9 \ln(2/\delta)}{2 n_t}} \right) \qquad (9)$$

*where $\mu_1, \ldots, \mu_T$ are probability measures, $\mathcal{Y} \subset \mathbb{R}^n$ is the random set obtained by $\mathcal{Y} = \{(f_t (\mathbf{x}_i^t)) : f_t \in \mathcal{F}\}$, and $G$ is Gaussian complexity.*

*Proof.* First, we consider a special case, let $T = 1$, we have $\mathbb{E}_{\mathbf{X} \sim \mu_t}[f_t(\mathbf{X})] - \sum_i \frac{1}{n_t} f_t(\mathbf{x}_i^t) \leq \frac{\sqrt{2\pi} G(\mathcal{Y})}{n_t} + \sqrt{\frac{9 \ln(2/\delta)}{2 n_t}}$ following (Maurer et al., 2016). Next, we generalize it and perform the summation operation with respect to $t$. $\qquad \square$

Lemma 1 yields that the task-averaged estimation error is bounded by the Gaussian complexity in multi-task learning. Next, we can move to the second step and then derive the hypothesis $g$-related generalization error bound for long-tail on graphs. We will give the key property of the Gaussian averages of a Lipschitz image in Lemma 2.

**Lemma 2** (Property of Gaussian Complexity (Maurer et al., 2016)). *Suppose $\mathcal{Y} \subseteq \mathbb{R}^n$ and $\phi : \mathcal{Y} \to \mathbb{R}^m$ is (Euclidean) Lipschitz continuous with Lipschitz constant $L$, we have*

$$G(\phi(\mathcal{Y})) \leq L G(\mathcal{Y}) \qquad (10)$$

From Lemma 2, we obtain the hypothesis $g$-related bound (S2) in Eq. 15. Next, we move to the third step: derived the generalization bound related to $h$ and $f_1, \ldots, f_T$, according to the chain rule of gaussian complexity presented in Lemma 3.

**Lemma 3** (Chain Rule of Gaussian Complexity). *Suppose we have $\mathcal{Y} \subseteq \mathbb{R}^n$ with (Euclidean) diameter $D(\mathcal{Y})$. $\mathcal{F}$ is a class of functions $f : \mathcal{Y} \to \mathbb{R}^m$, all of which have Lipschitz constant at most $L(\mathcal{F})$. Then, for any $y_0 \in \mathcal{Y}$,*

$$G(\mathcal{F}(\mathcal{Y})) \leq c_1 L(\mathcal{F}) G(\mathcal{Y}) + c_2 D(\mathcal{Y}) \mathtt{Range}(f_1, \ldots, f_T) + G \left( \mathcal{F} \left( y_0 \right) \right),$$

*where $c_1$ and $c_2$ are universal constants.*

*Proof.* Let

$$R(\mathcal{F}) = \sup_{\mathbf{y}, \mathbf{y}' \in \mathcal{Y}, \mathbf{y} \neq \mathbf{y}'} \mathbb{E} \sup_{f \in \mathcal{F}} \frac{\langle \gamma, l \left( f(\mathbf{y}) - f(\mathbf{y}') \right) \rangle}{\| \mathbf{y} - \mathbf{y}' \|}. \qquad (11)$$

where $\gamma$ is a vector of independent standard normal variables. Then following the definition of Rademacher complexity and the chain rule given in (Maurer et al., 2016), we have

$$G(\mathcal{F}(\mathcal{Y})) \leq c_1 L(\mathcal{F}) G(\mathcal{Y}) + c_2 D(\mathcal{Y}) R(\mathcal{F}) + G \left( \mathcal{F} \left( y_0 \right) \right) \qquad (12)$$

where $c_1$ and $c_2$ are constants. Furthermore,

$$\begin{aligned}
&\sup_{\mathbf{y}, \mathbf{y}' \in \mathcal{Y}, \mathbf{y} \neq \mathbf{y}'} \mathbb{E} \sup_{f \in \mathcal{F}} \frac{\langle \gamma, l \left( f(\mathbf{y}) - f(\mathbf{y}') \right) \rangle}{\| \mathbf{y} - \mathbf{y}' \|} \\
&\leq \sup_{\mathbf{y}, \mathbf{y}' \in \mathcal{Y}, \mathbf{y} \neq \mathbf{y}'} \mathbb{E} \left[ \sup_{f \in \mathcal{F}} \langle \gamma, l \left( f(\mathbf{y}) - y \right) \rangle - \sup_{f \in \mathcal{F}} \langle \gamma, l \left( f(\mathbf{y}') - y' \right) \rangle \right] \\
&\leq \sup_{\mathbf{y}, \mathbf{y}' \in \mathcal{Y}, \mathbf{y} \neq \mathbf{y}'} \left[ \frac{1}{n} \sum l(f(h(\mathbf{X})), \mathbf{y}) - \frac{1}{n} \sum l(f(h(\mathbf{X}')), \mathbf{y}') \right] \\
&\leq \max_t \frac{1}{n_t} \sum_{i=1}^{n_t} l(f_t(h(\mathbf{x}_i^t)), y_i^t) - \min_t \frac{1}{n_t} \sum_{i=1}^{n_t} l(f_t(h(\mathbf{x}_i^t)), y_i^t)
\end{aligned} \qquad (13)$$

$\qquad \square$

Finally, the generalization error bound under the setting of long-tail categories on graphs is given as in the following Theorem 1.

**Theorem 1** (Generalization Error Bound). *Given the node embedding extraction function $h \in \mathcal{H}$ and the task-specific classifier $f_1, \ldots, f_T \in \mathcal{F}$, with probability at least $1 - \delta, \delta \in [0, 1]$, we have*

$$\mathcal{E} - \hat{\mathcal{E}} \leq \sum_t \left( \frac{c_1 L G(\mathcal{H}(\mathbf{X}))}{n_t} + \frac{c_2 \sup_{h \in \mathcal{H}} \|h(\mathbf{X})\| Range(f_1, \ldots, f_T)}{n_t} + \sqrt{\frac{9 \ln(2/\delta)}{2n_t}} \right), \quad (3)$$

*where $\mathbf{X}$ is the node feature, $T$ is the number of tasks, $n_t$ is the number of nodes in task $t$, $L$ is Lipschitz constant, $G$ is Gaussian complexity, and $c_1$ and $c_2$ are universal constants.*

*Proof.* By lemma 1, we have that

$$\mathcal{E} - \hat{\mathcal{E}} \leq \sum_t \left( \frac{\sqrt{2\pi} G(S)}{n_t} + \sqrt{\frac{9 \ln(2/\delta)}{2n_t}} \right) \quad (14)$$

where $S = \{(l(f_t(h(X_i^t)), Y_i^t)) : f_t \in \mathcal{F} \text{ and } h \in \mathcal{H}\} \subseteq \mathbb{R}^n$. By the Lipschitz property of the loss function $l(\cdot, \cdot)$ and the contraction lemma 2, we have $G(S) \leq G(S')$, where $S' = \{(f_t(h(X_i^t))) : f_t \in \mathcal{F}$ and $h \in \mathcal{H}\} \subseteq \mathbb{R}^n$. Then

$$\mathcal{E} - \hat{\mathcal{E}} \leq \sum_t \left( \frac{\sqrt{2\pi} G(S')}{n_t} + \sqrt{\frac{9 \ln(2/\delta)}{2n_t}} \right). \quad (15)$$

Recall that $\mathcal{H}(\mathbf{X}) \subseteq \mathbb{R}^{Kn}$ is defined by

$$\mathcal{H}(\mathbf{X}) = \left\{ \left( h_k(X_i^t) \right) : h \in \mathcal{H} \right\}, \quad (16)$$

and define a class of functions $\mathcal{F}' : \mathbb{R}^{Kn} \to \mathbb{R}^n$ by

$$\mathcal{F}' = \left\{ y \in \mathbb{R}^{Kn} \mapsto \left( f_t(y_i^t) \right) : f_1, \ldots, f_T \in \mathcal{F} \right\}. \quad (17)$$

We have $S' = \mathcal{F}'(\mathcal{H}(\mathbf{X}))$. By Lemma 3 for universal constants $c_1'$ and $c_2'$

$$G(S') \leq c_1' L(\mathcal{F}') G(\mathcal{H}(\mathbf{X})) + c_2' D(\mathcal{H}(\mathbf{X})) \text{Range}(f_1', \ldots, f_T') + \min_{y \in Y} G(\mathcal{F}(y)). \quad (18)$$

We now bound the individual terms on the right-hand side above. Let $y, y' \in \mathbb{R}^{Kn}$, where $y = \{y_i^t\}$ with $y_i^t \in \mathbb{R}^K$ and $y' = \{y_i^{t\prime}\}$ with $y_i^{t\prime} \in \mathbb{R}^K$. Then for $f_1, \ldots, f_T \in \mathcal{F}$

$$\begin{aligned}
\|f(y) - f(y')\|^2 &= \sum \left( f_t(y_i^t) - f_t(y_i^{t\prime}) \right)^2 \\
&\leq L^2 \sum \|y_i^t - y_i^{t\prime}\|^2 = L^2 \|y - y'\|^2
\end{aligned} \quad (19)$$

so that $L(\mathcal{F}') \leq L$. Next, we take $y_0 = 0$ and the last term in (18) vanishes because we have $f(0) = 0$ for all $f \in \mathcal{F}$. Substitution in (18) and using $G(S) \leq G(S')$, we have

$$G(S) \leq c_1' L G(\mathcal{H}(\mathbf{X})) + c_2' \sqrt{T} D(\mathcal{H}(\mathbf{X})) \text{Range}(f_1, \ldots, f_T). \quad (20)$$

Finally, we bound $D(\mathcal{H}(\mathbf{X})) \leq 2 \sup_h \|h(\mathbf{X})\|$ and substitution in (14), the proof is completed. $\quad \square$

Theorem 1 shows that the generalization performance of long-tail categories on graphs can be improved by (1) reducing the loss range across all tasks $\text{Range}(f_1, \ldots, f_T)$, as well as (2) controlling the total number of tasks $T$. Below we give a corollary to theoretically explain how the two modules in TAIL2LEARN works.

**Corollary 1** (Effectiveness of TAIL2LEARN). *After the $l^{th}$ hierarchical task grouping layer, we group nodes into $T^{(l)}$ tasks with $T^{(l)} < T$, where $T$ is the number of classes. In addition, we can learn the task-specific predictors $f_1^{(l)}, \ldots, f_T^{(l)}$ with $\text{Range}(f_1^{(l)}, \ldots, f_T^{(l)}) < \text{Range}(f_1, \ldots, f_T)$*

*by long-tail balanced contrastive learning. Then we have that the upper bound of the error for* TAIL2LEARN *with l layers is smaller than the upper bound without the two modules, i.e.,*

$$
\sum_t \left( \frac{c_1 LG(\mathcal{H}(\mathbf{X}))}{n_t^{(l)}} + \frac{c_2 \sup_{h \in \mathcal{H}} \|h(\mathbf{X})\| Range(f_1^{(l)}, \ldots, f_T^{(l)})}{n_t^{(l)}} + \sqrt{\frac{9 \ln(2/\delta)}{2n_t^{(l)}}} \right)
$$
$$
\leq \sum_t \left( \frac{c_1 LG(\mathcal{H}(\mathbf{X}))}{n_t} + \frac{c_2 \sup_{h \in \mathcal{H}} \|h(\mathbf{X})\| Range(f_1, \ldots, f_T)}{n_t} + \sqrt{\frac{9 \ln(2/\delta)}{2n_t}} \right),
$$

(8)

*where $h \in \mathcal{H}$ is the shared GCN layer for representation learning. The number of instances in the $t^{th}$ class for layer $l$ is $n_t^{(l)}$, we have $\sum_t n_t^{(l)} = \sum_t n_t = n$, where $n$ is the total number of instances.*

*Proof.* Since we have $\texttt{Range}(\mathcal{F}^{(l)}) < \texttt{Range}(\mathcal{F})$, to compare the two upper bounds, we only need to compare the relationship between $\sum_t \frac{1}{n_t^{(l)}}$ and $\sum_t \frac{1}{n_t}$. Consider a special case where we group all nodes to the one hypertask in the $l^{th}$ layer, then we have one hypertask in layer $(l)$ and the number of nodes contained in this hypertask is $n_1 + \cdots + n_T$. While the number of nodes in each class $t$ is $n_t$, $t = 1, \cdots, T$. According to the relationship between the reconciled mean and the arithmetic mean, we have

$$
\frac{1}{n_1 + n_2 + \cdots + n_T} \leq \frac{T^2}{n_1 + n_2 + \cdots + n_T}
$$
$$
\leq \frac{1}{n_1} + \frac{1}{n_2} + \cdots + \frac{1}{n_T}.
$$

(21)

Without loss of generality, we have $\sum_t \frac{1}{n_t^{(l)}} \leq \sum_t \frac{1}{n_t}$, the proof is completed. $\qquad \square$

## D  OPTIMIZATION AND PSEUDO-CODE

Overall, the goal of the training process is to minimize the node classification loss (for few-shot annotated data), the unsupervised balanced contrastive loss (for task combiations in each layer), and the supervised contrastive loss (for categories). The node classification loss is defined as follows:

$$
\mathcal{L}_{NC} = \sum_{i=1}^{T} \mathcal{L}_{CE} \left( g(\mathcal{G}), \mathcal{Y} \right)
$$

(22)

where $\mathcal{L}_{CE}$ is the cross-entropy loss, $\mathcal{G}$ represents the input graph with few-shot labeled nodes and $\mathcal{Y}$ represents the labels. Then the overall loss function can be written as follows:

$$
\mathcal{L}_{total} = \mathcal{L}_{NC} + \gamma * (\mathcal{L}_{BCL} + \mathcal{L}_{SCL})
$$

(23)

where $\gamma$ balances the contribution of the three terms.

The pseudo-code of TAIL2LEARN is provided in Algorithm 1. Given an input graph $\mathcal{G}$ with few-shot label information $\mathcal{Y}$, our proposed TAIL2LEARN framework aims to predict $\hat{\mathcal{Y}}$ of unlabeled nodes in graph $\mathcal{G}$. We initialize all the task grouping, the unpooling layers and the classifier in Step 1. Steps 4-6 correspond to the task grouping process: We generate down-sampling graphs and compute node representations using GCNs. Then Steps 7-9 correspond to the unpooling process: We restore the original graph resolutions and compute node representations using GCNs. An MLP is followed for computing predictions after skip-connections between the task grouping and unpooling layers in Step 10. Finally, in Step 11, models are trained by minimizing the objective function. In Steps 13, we return predicted labels $\hat{\mathcal{Y}}$ in the graph $\mathcal{G}$ based on the trained classifier.

## E  DETAILS OF EXPERIMENT SETUP

### E.1  DATASETS

In this subsection, we give further details and descriptions on the six datasets to supplement Sec. 4.1. (1) Cora-Full is a citation network dataset. Each node represents a paper with a sparse bag-of-words

---

**Algorithm 1** The TAIL2LEARN Learning Framework.

---

**Require:**
  an input graph $\mathcal{G} = (\mathcal{V}, \mathcal{E}, \mathbf{X})$ with small node class long-tail ratio $\texttt{Ratio}_{LT}(\alpha)$ and few-shot annotated data $\mathcal{Y}$.

**Ensure:**
  Accurate predictions $\hat{\mathcal{Y}}$ of unlabeled nodes in the graph $\mathcal{G}$

 1: Initialize GCNs for graph embedding layer, task grouping layers, and unpooling layers; the MLP for the node classification task in $\mathcal{G}$.

 2: **while** not converge **do**

 3:  Compute node representations in a low-dimensional space of $\mathcal{G}$ via GCN for graph embedding layer.

 4:  **for** layer $l \in \{1, \dots, L\}$ **do**

 5:   Generate a down-sampling new graph (Eq. equation 4) and compute node representations for the new graph by $l^{\text{th}}$ task grouping layer.

 6:  **end for**

 7:  **for** layer $l \in \{1, \dots, L\}$ **do**

 8:   Restore the original graph resolutions (Eq. equation 5) and compute node representations for the origin graph by $l^{\text{th}}$ unpooling layer.

 9:  **end for**

10:  Perform skip-connections between the task grouping and unpooling layers, and calculate final node embeddings by feature addition. Employ an MLP layer for final predictions.

11:  Calculate node classification loss $\mathcal{L}_{NC}$ (Eq. equation 22) with node embeddings obtained in Step 3, calculate balanced contrastive loss $\mathcal{L}_{BCL}$ (Eq. equation 7) with node embeddings obtained in Step 5, and calculate supervised contrastive loss $\mathcal{L}_{SCL}$ (Eq. equation 6) with node embeddings obtained in Step 10. Update the hidden parameters of GCNs and MLP by minimizing the loss function in Eq. equation 23.

12: **end while**

13: return predicted labels $\hat{\mathcal{Y}}$ for unlabeled nodes in the graph $\mathcal{G}$.

---

vector as the node attribute. The edge represents the citation relationships between two corresponding papers, and the node category represents the research topic. (2) BlogCatalog is a social network dataset with each node representing a blogger and each edge representing the friendship between bloggers. The node attributes are generated from Deepwalk following (Perozzi et al., 2014). (3) Email is a network constructed from email exchanges in a research institution, where each node represents a member, and each edge represents the email communication between institution members. (4) Wiki is a network dataset of Wikipedia pages, with each node representing a page and each edge denoting the hyperlink between pages. (5) Amazon-Clothing is a product network which contains products in "Clothing, Shoes and Jewelry" on Amazon, where each node represents a product, and is labeled with low-level product categories for classification. The node attributes are constructed based on the product's description, and the edges are established based on their substitutable relationship ("also viewed"). (6) Amazon-Electronics is another product network constructed from products in "Electronics" with nodes, attributes, and labels constructed in the same way. Differently, the edges are created with the complementary relationship ("bought together") between products.

For additional processing, the first four datasets are randomly sampled according to train/valid/test ratios = 1:1:8 for each category. For the last two datasets, nodes are removed until the category distribution follows a long-tail distribution (here we make the head 20% categories containing 80% of the total nodes) with keeping the connections between the remaining nodes. We sort the categories by the number of nodes they contain and then downsample them according to Pareto distribution. When eliminating nodes, we remove nodes with low degrees and their corresponding edges. After semi-synthetic processing, the long-tailedness ratio of order 0.8 ($\texttt{Ratio}_{LT}(0.8)$) of train set is approximately equal to 0.25. For valid/test sets, we sample 25/55 nodes from each category. Notably, for Amazon-Clothing and Amazon-Electronics, we keep the same number of nodes for each category as test instances, so the values of bAcc and Acc are the same. To sum up, TAIL2LEARN is evaluated based on four natural datasets, and two additional datasets with semi-synthetic long-tail settings.

The statistics, the original class-imbalance ratio, and original long-tailedness ratio ($\texttt{Ratio}_{LT}(0.8)$ as defined in Definition 1) of each dataset are summarized in Table 5.

Table 5: Dataset statistics.

| Dataset | #Nodes | #Edges | #Attributes | #Classes | Imb. | $\text{Ratio}_{LT}$ |
|---|---|---|---|---|---|---|
| Cora-Full | 19,793 | 146,635 | 8,710 | 70 | 0.016 | 1.09 |
| BlogCatalog | 10,312 | 333,983 | 64 | 38 | 0.002 | 0.77 |
| Email | 1,005 | 25,571 | 128 | 42 | 0.009 | 0.79 |
| Wiki | 2,405 | 25,597 | 4,973 | 17 | 0.022 | 1.00 |
| Amazon-Clothing | 24,919 | 91,680 | 9,034 | 77 | 0.097 | 1.23 |
| Amazon-Electronics | 42,318 | 43,556 | 8,669 | 167 | 0.107 | 1.67 |

## E.2 BASELINES

Next, we describe each baseline in more details to supplement Sec. 4.1.

Classical long-tail learning methods: Origin utilizes a GCN (Kipf & Welling, 2017) as the encoder and an MLP as the classifier. Over-sampling (Chawla, 2003) duplicates the nodes of tail classes and creates a new adjacency matrix with the connectivity of the oversampled nodes. Reweighting (Yuan & Ma, 2012) penalizes the tail nodes to compensate for the dominance of the head nodes. SMOTE (Chawla et al., 2002) generates synthetic nodes by feature interpolation tail nodes with their nearest and assigns the edges according to their neighbors' edges. Embed-SMOTE (Ando & Huang, 2017) performs SMOTE in the embedding space instead of the feature space.

GNN-based long-tail learning methods: GraphSMOTE (Zhao et al., 2021) extends classical SMOTE to graph data by interpolating node embeddings and connecting the generated nodes via a pre-trained edge generator. It has two variants: GraphSMOTE$_T$ and GraphSMOTE$_O$, depending on whether the predicted edges are discrete or continuous. GraphMixup (Wu et al., 2021) performs semantic feature mixup and contextual edge mixup to capture graph feature and structure and then develops a reinforcement mixup to determine the oversampling ratio for tail classes. ImGAGN (Qu et al., 2021) is an adversarial-based method that uses a generator to simulate minority nodes and a discriminator to discriminate between real and fake nodes. GraphENS (Park et al., 2022) is an augmentation method, synthesizing an ego network for nodes in the minority classes with neighbor sampling and saliency-based node mixing. LTE4G (Yun et al., 2022) splits the nodes into four balanced subsets considering class and degree long-tail distributions. Then, it trains an expert for each balanced subset and employs knowledge distillation to obtain the head student and tail student for further classification.

## E.3 PARAMETER SETTINGS

For a fair comparison, we use vanilla GCN as backbone and set the hidden layer dimensions of all GCNs in baselines and TAIL2LEARN to 128 for Cora-Full, Amazon-Clothing, Amazon-Electronics and 64 for BlogCatalog, Email, Wiki. We use Adam (Kingma & Ba, 2015) optimizer with learning rate 0.01 and weight deacy $5e-4$ for all models. For the oversampling-based baselines, the number of imbalanced classes is set to be the same as in (Yun et al., 2022). And the scale of upsampling is set to 1.0 as in (Yun et al., 2022), that is, the same number of nodes are oversampled for each tail category. For GraphSMOTE, we set the weight of edge reconstruction loss to $1e-6$ as in the original paper (Zhao et al., 2021). For GraphMixup, we use the same default hyperparameter values as in the original paper (Wu et al., 2021) except settings of maximum epoch and Adam. For GraphENS (Park et al., 2022) and LTE4G (Yun et al., 2022), we adopt the best hyperparameter settings reported in the paper. For our model, the weight $\gamma$ of contrastive loss is selected in $\{0.01, 0.1\}$, the temperature $\tau$ of contrastive learning is selected in $\{0.01, 0.1, 1.0\}$. We set the depth of the hierarchical graph neural network to 3; node embeddings are calculated for the first layer, the number of tasks is set to the number of categories for the second layer, and the number of tasks is half the number of categories for the third layer. In addition, the maximum training epoch for all the models is set to $10,000$. If there is no additional setting in the original papers, we set the early stop epoch to $1,000$, i.e., the training stops early if the model performance does not improve in 1000 epochs. All the experiments are conducted on an A100 SXM4 80GB GPU.

# F  PARAMETER AND COMPLEXITY ANALYSIS

**Hyperparameter Analysis:** We study seven hyperparameters of TAIL2LEARN: (1) the weight $\gamma$ to balance the contribution of three losses; (2) the temperature $\tau$ of balanced contrastive loss in M1; (3) use first-order GCN or vanilla GCN; (4) the activation function in GCN; (5) the number of hidden dimensions; (6) the dropout rate; and (7) the structure of hierarchical graph neural network including the depth and the number of tasks. First we show the sensitivity analysis with respect to weight $\gamma$ and temperature $\tau$, and the results are shown in Figure 5. The fluctuation of the bAcc (z-axis) is less than 5%. The bAcc is slightly lower when both weight $\gamma$ and temperature $\tau$ become larger. The analysis results for the remaining five hyperparameters are presented in Figure 6. For analyzing these hyperparameters, all the experiments are conducted with weight $\gamma = 0.01$ and temperature $\tau = 0.01$. For hierarchical structure, we investigated different hierarchical depths and task sizes: (1) using node embeddings and prototype embeddings for 70 (# of classes) tasks; (2) using node embeddings, prototype embeddings for 198 tasks, and prototype embeddings for 70 hypertasks; (3) using node embeddings, prototype embeddings for 70 tasks, and prototype embeddings for 35 hypertasks. Overall, we find TAIL2LEARN is reliable and not sensitive to the hyperparameters under a wide range.

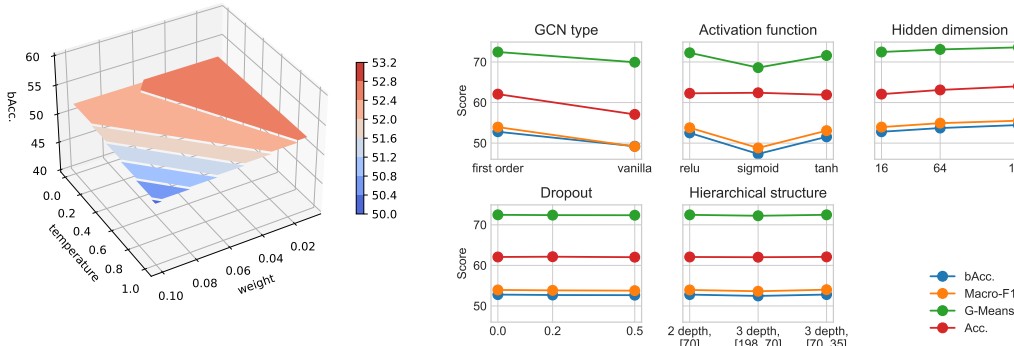

Figure 5: Hyperparameter analysis on Cora-Full with respect to weight $\gamma$ and temperature $\tau$.

Figure 6: Hyperparameter analysis on Cora-Full.

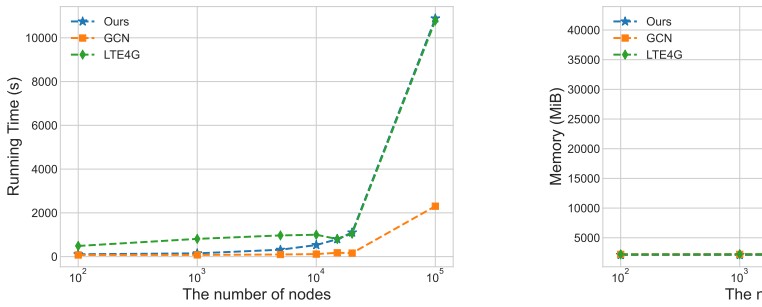

Figure 7: Time complexity analysis w.r.t. the number of nodes.

Figure 8: Space omplexity analysis w.r.t. the number of nodes.

**Complexity analysis:** We report the running time and memory usage of TAIL2LEARN, GCN, and LTE4G (a efficient state-of-the-art method). For better visualization, we conduct experiments on synthetic datasets with an increasing graph size, i.e., from 100 to 100,000 nodes. As depicted in Figure 7, our approach TAIL2LEARN consistently exhibits superior or similar running time compared to the LTE4G method. Although our method has slightly higher running time than GCN, the gap between our approach and GCN remains modest especially when for graph sizes smaller than

$10^4$. The relationship between the running time of our model and the number of nodes is similarly linear. The best space complexity of our method can reach $O(nd + d^2 + |\mathcal{E}|)$, which is linear in the number of nodes and the number of edges. From the memory usage given in Figure 8, it is shown that TAIL2LEARN exhibits significantly superior memory usage compared to LTE4G and closely approximates the memory usage of GCN. The results illustrate the scalability of our method.

