# OpenReview forum: "Characterizing Long-Tail Categories on Graphs via A Theory-Driven Framework"
_ICLR.cc/2024/Conference — ICLR 2024 Conference Withdrawn Submission_

### Official Review · Reviewer_pu5a · 2023-10-25

**Soundness:** 2 fair
**Presentation:** 3 good
**Contribution:** 2 fair
**Rating:** 5
**Confidence:** 4

**Summary:**

The paper confronts a significant challenge in long-tailed classification on graphs. While most prior research concentrates on mitigating bias, this paper offers a fresh perspective by introducing a theoretical framework for characterizing long-tail categories and improving generalization in real-world scenarios. The authors present the TAIL2LEARN framework, encompassing hierarchical task grouping and long-tailed balanced contrastive learning. Notably, the experimental results demonstrate promising performance, outperforming state-of-the-art methods.

**Strengths:**

- The proposed approach is novel and addresses a significant gap in the existing literature by providing a theoretical foundation for long-tail classification on graphs. The motivation for this work is well-defined and highlights the need for a deeper understanding of class imbalances and generalization performance.
- A notable strength of the paper is its comprehensive theoretical analysis, which includes the development of a Generalization Error Bound that substantiates the effectiveness of the proposed method.
- The experimental results effectively illustrate the superiority of the proposed TAIL2LEARN framework. By showcasing its effectiveness in characterizing long-tail categories on real-world graph datasets, the authors provide practical evidence of their method's capabilities.

**Weaknesses:**

- One potential weakness of the paper is that the hierarchical task grouping approach employed by the authors seems similar with existing techniques like Graph U-Net [1]. Although the authors have extended these prior methods to facilitate multi-task learning and task grouping with theoretical backing, it may require clarification about what sets the TAIL2LEARN framework apart from the existing Graph U-Net. Further clarification and a more detailed comparison between the two would be beneficial to better understand the novelty and differentiation of the proposed framework.
- While the authors have approached long-tailed classification as a multi-task learning problem, they have configured the number of tasks in the second layer to align with the number of categories. It might be worth considering whether the authors have explored the possibility of subdividing the samples into more finely-grained subclasses, which means increasing the number of tasks in the second layer beyond the number of categories.
- The authors claimed that $\mathcal{L}_{BCL} potentially controls the range of losses for different tasks. However, the paper lacks experimental results to support this claim, which could contribute to a more robust evaluation of the method's effectiveness.

[1] Gao H, Ji S. Graph u-nets[C]//international conference on machine learning. PMLR, 2019: 2083-2092.

**Questions:**

See above.

---

> ### Author Response · Authors · 2023-11-21
>
> >**Q1:** It may require clarification about what sets the TAIL2LEARN framework apart from the existing Graph U-Net.
>
> **A1:** We are the first to consider the long-tail problem in the task space instead of the instance space. The proposed theoretical analysis and framework consider the complexity of the task. In particular, our model is different from Graph U-Net in the following aspects:
>
> - one of our key ideas is to group similar tasks as a hypertask, while Graph U-Net samples a subset of important nodes ;
>
> - Graph U-Net keeps the indexes of selected nodes for unpooling operation, while our method preserves the tasks hierarchically information to capture the complex relationships among tasks.
>
> &nbsp;
> >**Q2:** It might be worth considering whether the authors have explored the possibility of subdividing the samples into more finely-grained subclasses.
>
> **A2:** Thank you for your insightful comment!
>
> - Yes, the samples can be finely-grained grouped by increasing the number of tasks. **There have been some work delved into overclustering**, i.e., the number of clusters is larger than the number of categories. The work [1,2] posits that overclustering can be beneficial for the model to learn expressive features.
>
> - We showed the impact of varying the number of tasks on model performance in Appendix F (last subplot of Figure 6). **We have also conducted additional experiments to explore the impact.** The experimental results show that our model can achieve great model performance when the hyperparameters are in a certain reasonable range. But there is a slight performance degradation when the number of hypertasks is small.
>
> |          |      | Cora_Full |         |      |
> |----------|------|-----------|---------|------|
> |          | bAcc | Macro-F1  | G-Means | Acc  |
> | [198,70] | 55.5 | 56.7      | 74.2    | 64.6 |
> | [70,35]  | 55.8 | 57.1      | 74.5    | 64.7 |
> | [2,1]    | 54.9 | 56.8      | 73.9    | 65.5 |
>
> `[1]` Ji, Xu, Andrea Vedaldi, and João F. Henriques. "Invariant information clustering for unsupervised image classification and segmentation. ICCV, 2019.
>
> `[2]` Kim, Yunji, and Jung-Woo Ha. "Contrastive fine-grained class clustering via generative adversarial networks." ICLR, 2022.
>
> &nbsp;
> >**Q3:** The authors claimed that $\mathcal{L}_{BCL} potentially controls the range of losses for different tasks. However, the paper lacks experimental results to support this claim.
>
> **A3:** The loss range is defined as $\max_{t}\frac{1}{n_t}\sum_{i=1}^{n_t}l(f_t(h(\mathbf{x}^t_{i})), y^t_{i}) - \min_{t}\frac{1}{n_t}\sum_{i=1}^{n_t}l(f_t(h(\mathbf{x}^t_{i})), y^t_{i})$. Intuitively, we assume that the model with $\mathcal{L}\_{BCL}$ can maintain good performance on head tasks, so the second term of the range almost remains constant; however, the balanced contrastive learning improves performance on the tail task, so the first term of the range decreases and therefore can help reduce the loss range. Moreover, in the Table below we empirically show that using $\mathcal{L}\_{BCL}$ improves the model performance from 54.6 to 55.8 (bAcc) on the Cora_Full dataset, which also demonstrates the efficacy of our balanced contrastive learning.
>
> |M1|$\mathcal{L}_{BCL}$|$\mathcal{L}_{SCL}$|$\mathcal{L}_{CE}$|bAcc.|Macro-F1|G-Means|Acc.|
> |-|-|-|-|-|-|-|-|
> |&check;|&check;|&check;|&check;|$\textbf{55.8}\pm0.5$|$\textbf{57.1}\pm0.5$|$\textbf{74.5}\pm0.3$|$\textbf{64.7}\pm0.7$|
> |&check;||&check;|&check;|$54.6\pm0.4$|$56.2\pm0.4$|$73.7\pm0.3$|$64.3\pm0.4$|
> ||||&check;|$52.8\pm0.6$|$54.5\pm0.7$|$72.5\pm0.4$|$62.7\pm0.5$|

---

> > ### Comment · Reviewer_pu5a · 2023-11-22
> >
> > The author has partially addressed my concerns and I will keep my score unchanged.

---

### Official Review · Reviewer_Wqoo · 2023-10-31

**Soundness:** 1 poor
**Presentation:** 2 fair
**Contribution:** 1 poor
**Rating:** 3
**Confidence:** 4

**Summary:**

This paper investigate long-tailed classification on graphs by providing a PAC generalization bound in a multi-task learning fashion, which is characterized by the task number and overall loss range. As a solution, the authors propose Tail2Learn, a learning framework for long-tailed node classification, which reduces task complexity by hierarchically grouping tasks and adopting a contrastive loss to adaptively balance the gradients of both head and tail classes to control the loss range.

**Strengths:**

+ The method presented in this paper is straightforward and easily comprehensible.

+ The approach of addressing the long-tail problem through a multi-task learning perspective appears to be original.

+ The empirical results in the experiment section indicate a promising improvement compared to the baseline methods.

**Weaknesses:**

- The theoretical aspect of the paper appears to be quite preliminary, lacking in-depth analysis and original contributions. It appears to heavily rely on Theorem 8 from a previous work [1]. Additionally, some statements and derivations are unclear and contain errors, making it challenging to verify their correctness. For specific concerns, please refer to the detailed questions.

- While the shift in perspective towards multi-task learning is novel, the proposed method is essentially a combination of existing, well-known techniques, such as hierarchy graph pooling [2] and contrastive loss [3].

- The discussion and comparison of some relevant work (such as [4][5]) are missing in this paper.

[1] Maurer et al., The Benefit of Multitask Representation Learning, 2016

[2] Ying et al., Hierarchical graph representation learning with differentiable pooling, 2018

[3] Zheng et al., Tackling Oversmoothing of GNNs with Contrastive Learning, 2021

[4] Zhang et al., "Graph-less Neural Networks: Teaching Old MLPs New Tricks Via Distillation, 2022

[5] Zheng et al., Cold Brew: Distilling Graph Node Representations with Incomplete or Missing Neighborhood, 2022

**Questions:**

1. In Lemma 1, after applying the PCA bound provided by [1], how is the normalization term $1/T$ eliminated? If $1/T$ needs to remain, I would question one of the main claims in the abstract: "generalization performance of long-tailed classification is dominated by the total number of tasks'' (as stated in the abstract). This is because, after accounting for $1/T$, such a bound may no longer scale with the number of tasks. Actually, Theorem 2 in [1] even suggests the generalization error decays with $O(1/\sqrt{T})$ when transferred to new task.

2. In Lemma 3, why is the definition of $R(F)$ (Eq. 11) different from Eq. 4 in [1]? Note that Eq. 4 has $f(y) - f(y')$ in the numerator, while Eq. 11 has $l(f(y) - f(y'))$ in the numerator.

3. In the proof of Corollary 1, it remains unclear to me how the inequality between the last terms (under the square root) is established.

4. Why is this analysis focused specifically on long-tail classification on graphs? Can it be extended to the general long-tail learning problem?

5. Can the authors explain why the proposed approach, Tail2Learn, takes on the form of $f \circ h," resembling a general multi-task learning framework?

6. It is known that task complexity is not always harmful. Instead, improving task diversity can be helpful for multi-task learning [2][3]. Does this contradict to this paper’s claim?

[1] Maurer et al., The Benefit of Multitask Representation Learning, 2016

[2] Tripuraneni et al., On the Theory of Transfer Learning: The Importance of Task Diversity, 2020

[3] Du et al., Few-Shot Learning via Learning the Representation, Provably, 2020

---

> ### Author Response · Authors · 2023-11-21
>
> **Thank you for your careful review; it is very helpful for enhancing the quality of our paper. Yes, Theorem 1 should retain the normalization term 1/T. We are still revising our paper to accommodate this change.**
>
> In addition, we think that **'improving task diversity can be helpful for multi-task learning' does not contradict our paper.** While our paper incorporates the paradigm of multi-task learning, it is crucial to highlight the fundamental differences between long-tail learning and multi-task learning: long-tail learning not only involves a large number of classes but also exhibits substantial class-membership imbalance.
>
> - In the context of **multi-task learning** discussed in previous work `[1]`, each task corresponds to $n$ observed input samples, where enhancing task diversity proves beneficial for performance. Increasing the number of tasks $T$ decreases the cost associated with estimating the representation $h$, while increasing the number of samples $n$ decreases the cost of estimating task-specific predictors.
>
> - On the other hand, in **long-tail learning**, where we define task complexity in situations where a vast majority of samples belong to a few head classes, and a minority of samples are allocated to tail classes, increasing the number of classes  (particularly tail classes) may increase the difficulty of learning a salient representation and depicting the support regions of each classes. In long-tail learning, task complexity is relevant to $n_1,\ldots , n_T, T$, thus task complexity does not reduce the generalization error.
>
> - Our **experimental results** show great performance in utilizing hierarchical grouping (Table 1), especially in the harsh long-tail setting (Table 2).
>
> Further, we argue that **'in the long-tail setting, the increase in task complexity, especially considering tail categories have very few samples, raises the difficulty in correct classification'.** This is because the shared representation $h$ can be considered as a consensus of different views. The cost of maximizing consistency and learning a uniform representation from multiple subspaces will increase when categories (especially tail categories) increase. And when the number of categories increases, the loss range remains constant or increases.
>
> `[1]` Maurer et al., The Benefit of Multitask Representation Learning, 2016.

---

### Official Review · Reviewer_Td6K · 2023-10-31

**Soundness:** 2 fair
**Presentation:** 2 fair
**Contribution:** 2 fair
**Rating:** 3
**Confidence:** 3

**Summary:**

The paper studies long-tail categories in graphs. It proposes a generalization bound for long-tail classification on graphs, as well as a method TAIL2LEARN for long-tailed classification on graphs. The method includes a hierarchical task grouping module to reduce complexity of task space and a contrastive learning module to balance the gradients of head and tail classes. Experiments are conducted to evaluate the method.

**Strengths:**

1. The paper provides theoretical studies and arrives at a generalization bound.

2. The paper presentation includes rich contents, with tables and figures well organized.

3. The conducted experiments look correct with ablation studies included and code provided.

**Weaknesses:**

1. Related works not well addressed. The long-tail categories studied in the paper is the same as the node-level imbalanced-class problem in graph. The imbalanced class problem has been studied intensively for graphs, which is closely related to this work but not sufficiently discussed in its related works. The paper lacks a thorough review of related literature. Some missing related works are [1-6].

2. Following the above point, the experiments should include some of the missing imbalanced class baselines.

3. The correctness of Corollary 1 is unclear. Why can contrastive learning guarantee to learn the predictors $f_1^{(l)}, . . . , f_T^{(l)}$ with $Range(f_1^{(l)}, . . . , f_T^{(l)}) < Range(f_1, . . . , f_T)$? In its proof, why do we only need to compare the relationship between $\sum _t 1/(n_t^{(l)})$ and  $\sum _t 1/(n_t)$? And how is the special case of all nodes in one hypertask generalized to prove $\sum _t 1/(n_t^{(l)})\leq \sum _t 1/(n_t)$? The proof should be clearly given step-by-step instead of ambiguously stated.



[1] Imgcl: Revisiting graph contrastive learning on imbalanced node classification

[2] Boosting-GNN: boosting algorithm for graph networks on imbalanced node classification

[3] Graph neural network with curriculum learning for imbalanced node classification

[4] Co-Modality Graph Contrastive Learning for Imbalanced Node Classification

[5] Diving into Unified Data-Model Sparsity for Class-Imbalanced Graph Representation Learning

[6] TAM: topology-aware margin loss for class-imbalanced node classification

**Questions:**

Please see Weaknesses.

---

> ### Author Response · Authors · 2023-11-21
>
> >**Q:** Related works not well addressed. The long-tail categories studied in the paper is the same as the node-level imbalanced-class problem in graph.
>
> **A**: Thank you for your suggestion!
>
> - We would like to respectfully emphasize, as we identified in the introduction, that there are three fundamental challenges of long-tail classification problems (C1. Highly-skewed data distribution, C2.Label scarcity, and C3. Task complexity). **While there exists some similarity between long-tail and imbalance classification, long-tail learning on the challenges arising from a significantly larger number of tail classes.** Consequently, these two problem domains are not the same.
>
> - We have cited work `[1-6]` for imbalance in the section of related work. Furthermore, we have introduced the TAM method as a baseline for our experiments. We will provide the complete experimental results in a later version.
>
> | Method |              | Email        |              |              | Wiki         |              |              |  |
> |--------|--------------|--------------|--------------|--------------|--------------|--------------|--------------|--------------|
> |        | bAcc         | Macro-F1     | G-Means      | Acc          | bAcc         | Macro-F1     | G-Means      | Acc          |
> | Origin | 48.9+4.5     | 45.2+4.3     | 69.5+3.2     | **66.7+2.1** | 48.2+1.5     | 49.9+1.9     | 68.6+1.1     | 64.2+0.9     |
> | TAM    | 48.9+4.6     | 40.3+3.5     | 69.5+3.3     | 54.6+4.3     | 49.3+2.1     | 49.1+1.6     | 69.3+1.5     | 62.9+1.5     |
> | Ours   | **50.5+3.0** | **46.6+3.0** | **70.7+2.1** | 65.4+1.7     | **52.8+2.0** | **54.1+2.3** | **71.9+1.4** | **67.2+1.1** |
>
> | Method |              | Amazon-      | Clothing     |              | Amazon-      | Electronics  |              |  |
> |--------|--------------|--------------|--------------|--------------|--------------|--------------|--------------|--------------|
> |        | bAcc         | Macro-F1     | G-Means      | Acc          | bAcc         | Macro-F1     | G-Means      | Acc          |
> | Origin | 9.9+0.2      | 9.5+0.2      | 31.3+0.3     | 9.9+0.2      |16.9+0.2     | 15.2+0.2     | 41.0+0.3     | 16.9+0.2     |
> | TAM    | 10.9+2.5     | 9.8+2.4      | 32.6+3.9     | 10.9+2.5     | 18.5+3.4     | 15.8+3.5     | 42.7+4.0     | 18.5+3.4     |
> | Ours   | **17.1+0.5** | **16.8+0.6** | **41.1+0.6** | **17.1+0.5** | **23.6+0.9** | **21.0+1.3** | **48.5+1.0** | **23.6+0.9** |

---

### Official Review · Reviewer_8rhB · 2023-11-01

**Soundness:** 4 excellent
**Presentation:** 4 excellent
**Contribution:** 3 good
**Rating:** 8
**Confidence:** 4

**Summary:**

While current methods focusing on the long-tail problem in graphs have shown notable improvements, this work, Tail2Learn, approaches from a different perspective and formulates the long-tail classification problem into a multi-task learning framework. Built upon theoretical findings, it controls the complexity of task space and the loss range of task-specific classifiers by offering remedies such as hierarchical task grouping and long-tailed balanced contrastive learning. The experiments on the node classification task show the efficacy of Tail2Learn in real-world long-tailed graph datasets.

**Strengths:**

1. I quite enjoyed reading this paper. Overall, the claims of this paper are well-formulated, and its remedies for the theoretical findings are well-supported.

2. The proposed Definition 1, Long-Tailedness Ratio, is intuitive and straightforward. This metric can be generalized to balanced cases, such as 5 classes having 20 training samples each, as it would have a value of 4 in the 80th percentile.  This contribution would further enrich the long-tail GNN community.

3. The empirical performance aligns with the theoretical motivation. Also, the paper is well-written and easy to follow.

**Weaknesses:**

*Major*
1. In M1. Hierarchical Task Grouping, I agree that this approach can reduce label scarcity and task complexity. However, I am concerned whether hierarchical grouping across different classes might compromise the distinctiveness of each class. That is, there could be a trade-off between achieving reduced complexity and maintaining distinctiveness among classes. Although there exists a module for contrastive loss between different classes, its contribution remains unclear. A more detailed discussion of such situations should be provided.

2. In M2. Long-Tail Balanced Contrastive Learning, the utilization of supervised contrastive loss seems reasonable. However, given the long-tail situation, there would be very few training samples with labels for tail classes. Consequently, the positive pairs within tail classes would be significantly fewer compared to the head classes. Can you elucidate how Tail2Learn can work effectively in this scenario?

3. Although the overall performance of Tail2Learn is effective in current datasets, can you provide more details about the improvements made in tail classes as shown in Figure 4 in LTE4G [1]? This would offer a more comprehensive understanding of Tail2Learn's efficacy in terms of improvement in tail classes without sacrificing performance in head classes.

4. Can Tail2Learn generalize well on graph datasets having a relatively small number of classes such as Cora, CiteSeer, and PubMed?

*Minor*
1. Although Definition 1, Long-Tailedness Ratio, is well-designed, for clarity, at first glance, I expected the semantic meaning to refer to "how severe the data distribution is long-tailed". However, in actuality, the semantic meaning is "the lower the severity of long-tailedness." Have you considered the reciprocal version of the current long-tailedness ratio?

2. The notation (e.g., subscripts) in Equation 6 and Equation 7 appears to be exactly the same, while the underlying meaning is different. For clarity, I suggest differentiating the notations that denote specific classes and specific hypertasks, as they do not necessarily have to be the same value.

3. The performance of ImGAGN [2] in Table 1 seems unusually low compared to classical GNN, although it is originally designed to alleviate class long-tailedness. Can you provide further explanations for this?

If the above concerns are properly addressed, I would be very happy to raise my score on the current rating.

[1] [CIKM 2022] LTE4G: Long-Tail Experts for Graph Neural Networks
[2] [KDD 2021] ImGAGN:Imbalanced Network Embedding via Generative Adversarial Graph Networks

**Questions:**

See the Weaknesses.

---

> ### Author Response · Authors · 2023-11-21
> **Part 1/2**
>
> >**Q1:** There could be a trade-off between achieving reduced complexity and maintaining distinctiveness among classes.
>
> **A1:** Thank you for your insightful comment!
>
> - Yes, there has been some work delved into overclustering, i.e., the number of clusters is larger than the number of categories. The work `[1,2]` posits that **overclustering can be beneficial for the model to learn expressive features.**
>
> - We showed the impact of varying the number of hypertasks on model performance in Appendix F (last subplot of Figure 6). We have also conducted additional experiments to explore the cases when the number of overtasks >, =, < the number of categories. The experimental results show our model (also using cross-entropy loss and contrastive loss) can ensure sufficient distinctiveness and achieve great model performance. But there is a slight performance degradation when the number of hypertasks is small (i.e., # of hypertasks is way smaller than the # of classes), potentially attributed to issues of indistinctiveness.
>
> |          |      | Cora_Full |         |      |
> |----------|------|-----------|---------|------|
> |          | bAcc | Macro-F1  | G-Means | Acc  |
> | [198,70] | 55.5 | 56.7      | 74.2    | 64.6 |
> | [70,35]  | 55.8 | 57.1      | 74.5    | 64.7 |
> | [2,1]    | 54.9 | 56.8      | 73.9    | 65.5 |
>
> `[1]` Ji, Xu, Andrea Vedaldi, and João F. Henriques. "Invariant information clustering for unsupervised image classification and segmentation. ICCV, 2019.
>
> `[2]` Kim, Yunji, and Jung-Woo Ha. "Contrastive fine-grained class clustering via generative adversarial networks." ICLR, 2022.
>
> &nbsp;
> >**Q2:** Given the long-tail situation, there would be very few training samples with labels for tail classes. Can you elucidate how Tail2Learn can work effectively in this scenario?
>
> **A2:** Yes, we use balanced contrast loss to mitigate this problem, which is shown as follows (Section 3.2 M2).
>
> $$\mathcal{L} _{SCL}(\mathbf{z} _i) = -\frac{1}{n _{t}-1} \times \sum _{j \ \in\mathcal{V} _t \backslash i} \log \frac{\exp \left(\mathbf{z} _{i} \cdot \mathbf{z} _{j}/\tau\right)}{\sum _{1\leq q\leq T} \frac{1}{n _{q}} \sum _{k\in\mathcal{V} _q} \exp \left(\mathbf{z} _{i} \cdot \mathbf{z} _{k}/\tau\right)}$$
>
> $$\mathcal{L}(\mathbf{z} _i)=-\frac{1}{n _{t}-1} \times \sum _{j \ \in\mathcal{V} _t\backslash i} \log \frac{\exp \left(\mathbf{z} _{i} \cdot \mathbf{z} _{j}/\tau\right)}{\sum _{k\in\mathcal{V}} \exp \left(\mathbf{z} _{i} \cdot \mathbf{z} _{k}/\tau\right)}$$
>
> To be noted that,
>
> - Since there are significant differences in the number of samples between the head and tail categories, the head categories have a great contribution to the denominator in classic contrastive loss. To address this issue, **our loss function incorporates a category-based normalization in the denominator, thereby mitigating the dominance of the head categories.**
>
> - In addition, we normalize by the number of positive pairs ($n_t-1$) to compute an average instead of a sum to alleviate the label imbalance issue.
>
> - When applying to hypertasks, the node set $\mathcal{V}_t$ includes both task nodes belonging to hypertask $t$ and **prototypes of hypertask $t$**, which increases the number of positive pairs, especially helping tail categories.
>
> &nbsp;
> >**Q3:** Can you provide more details about the improvements made in tail classes as shown in Figure 4 in LTE4G [1]?
>
> **A3:** Thanks for your valuable suggestion! **In Figure 1, we plot the model performance on each category, showing that our model outperforms the original GCN method.** In addition, we have added the following figure to provide more details (**https://anonymous.4open.science/r/Tail2Learn-CE08/figs/resultPerClass.pdf**).

---

> ### Author Response · Authors · 2023-11-21
> **Part 2/2**
>
> >**Q4:** Can Tail2Learn generalize well on graph datasets having a relatively small number of classes such as Cora, CiteSeer, and PubMed?
>
> **A4:**
> - The datasets Cora, Citeseer, and PubMed are widely used graph datasets. **However, they exhibit a relatively limited number of categories**, a relatively low degree of imbalance, and do not follow a long-tail distribution. Hence these datasets were initially excluded from our paper.
>
> - We have conducted our model on the datasets compared with baselines. The experimental results suggest the great performance of our model on these datasets. We will provide the complete results in the appendix of the later version.
>
> | Method        |              | PubMed       |              |              |              | Cora         |              |              |
> |---------------|--------------|--------------|--------------|--------------|--------------|--------------|--------------|--------------|
> |               | bAcc         | Macro-F1     | G-Means      | Acc          | bAcc         | Macro-F1     | G-Means      | Acc          |
> | Origin        | 84.5+0.2     | 84.6+0.2     | 88.3+0.2     | 85.1+0.2     | 78.5+1.0     | 80.0+0.7     | 87.1+0.6     | 81.6+0.6     |
> | Over-sampling | 83.3+0.4     | 83.7+0.4     | 87.4+0.3     | 84.4+0.4     | 77.2+1.0     | 79.2+0.7     | 86.4+0.6     | 81.0+0.7     |
> | Re-weight     | 84.5+0.2     | 84.6+0.2     | 88.3+0.2     | 85.1+0.2     | 79.0+0.7     | 80.1+0.8     | 87.4+0.5     | 81.7+0.7     |
> | SMOTE         | 83.5+0.5     | 83.9+0.4     | 87.6+0.4     | 84.5+0.4     | 76.7+1.1     | 78.8+0.8     | 86.0+0.7     | 80.7+0.7     |
> | Embed-SMOTE   | 84.5+0.3     | 84.6+0.2     | 88.3+0.2     | 85.1+0.2     | 78.9+0.9     | 80.0+0.7     | 87.3+0.5     | 81.6+0.6     |
> | GraphSMOTE_T  | 84.3+0.3     | 84.4+0.2     | 88.2+0.2     | 85.1+0.2     | 80.8+1.0     | 81.5+1.1     | 88.5+0.6     | 83.0+0.9     |
> | GraphSMOTE_O  | 84.3+0.2     | 84.3+0.3     | 88.1+0.2     | 85.0+0.3     | 80.5+1.1     | 81.3+1.3     | 88.3+0.7     | 82.8+1.1     |
> | LTE4G         | 85.5+0.2     | 84.9+0.2     | 89.0+0.1     | 85.4+0.2     | 81.4+0.8     | 81.3+0.9     | 88.9+0.5     | 83.0+0.9     |
> | Ours          | **85.8+0.3** | **86.0+0.2** | **89.6+0.2** | **86.6+0.2** | **81.4+0.6** | **82.3+0.6** | **88.9+0.4** | **83.6+0.6** |
>
> &nbsp;
> >**Q5:** Have you considered the reciprocal version of the current long-tailedness ratio?
>
> **A5:** Thanks for your suggestion! We will provide detailed discussions and data statistics by using the reciprocal of long-tailedness ratio in our later version.
>
> &nbsp;
> >**Q6:** The notation (e.g., subscripts) in Equation 6 and Equation 7 appears to be exactly the same, while the underlying meaning is different.
>
> **A6:** Thanks for your suggestion! **We try to use $\prime$ to distinguish classes and hypertasks.** For example, use $j \in\mathcal{V} _t\backslash i$ to represent all nodes belonging to category $t$ except $i$, and use $j \in\mathcal{V} _{t^\prime}\backslash i \cup \{z _{t^\prime}\}$ to define all nodes belonging to hypertask $t^\prime$ except $i$ include the prototype $z _{t^\prime}$ of hypertask $t^\prime$. We seek your perspective on the proposed change. We intend to apply this change in the later version if you find it is better for understanding.
>
> &nbsp;
> >**Q7:** The performance of ImGAGN [2] in Table 1 seems unusually low compared to classical GNN.
>
> **A7:** In the default setting of ImGAGN, **it considers only one class as the tail category and generates synthetic minority nodes.** Therefore, the performance of the method is relatively weak. We adjusted the number of categories considered as tail categories in ImGAGN, and the experimental results did show an improvement in performance.
>
> |                 |          | Cora_Full|          |          |
> |-----------------|----------|----------|----------|----------|
> |                 | bAcc     | Macro-F1 | G-Means  | Acc      |
> | ImGAGN (1 tail) | 9.3+1.1  | 6.6+1.0  | 30.2+1.9 | 20.9+2.1 |
> | ImGAGN (7 tail) | 15.8+1.6 | 12.4+1.6 | 39.5+2.1 | 32.2+2.7 |

---

> > ### Comment · Reviewer_8rhB · 2023-11-23
> >
> > I appreciate the author's efforts during the rebuttal process.
> > All of my concerns have been alleviated, and as a result, I will raiese my score to 8.